# Cellpose3: one-click image restoration for improved cellular segmentation

**Carsen Stringer** ✉ **& Marius Pachitariu** ✉

Generalist methods for cellular segmentation have good out-of-the-box performance on a variety of image types; however, existing methods struggle for images that are degraded by noise, blurring or undersampling, all of which are common in microscopy. We focused the development of Cellpose3 on addressing these cases and here we demonstrate substantial out-of-the-box gains in segmentation and image quality for noisy, blurry and undersampled images. Unlike previous approaches that train models to restore pixel values, we trained Cellpose3 to output images that are well segmented by a generalist segmentation model, while maintaining perceptual similarity to the target images. Furthermore, we trained the restoration models on a large, varied collection of datasets, thus ensuring good generalization to user images. We provide these tools as 'one-click' buttons inside the graphical interface of Cellpose as well as in the Cellpose API.

Microscopy techniques can be adapted to many different imaging problems, but there are multiple constraints that can limit the quality of the acquired images. For example, some biological samples may be damaged by illumination light[1,2] and some fluorescent sensors may bleach during timelapse microscopy[3]. Both these scenarios require reduced illumination, which increases the shot noise[4,5]. Opening up the aperture of a confocal or widefield microscope increases photons and reduces shot noise, but also results in blurring, which is a different type of degradation[6]. Microscopy in thick tissue may also generate blurry images as well as shot noise due to light scattering and aberrations[7,8]. Other samples may be so large that it would take an unreasonable amount of time to image them at high resolution and the volumes may be undersampled during acquisition[9,10]. Thus, images may be degraded by either shot noise, blurring or undersampling across many types of microscopy applications.

Cellular segmentation models like Cellpose do not work well out of the box on degraded images, likely because there are very few noisy images in the training datasets (Extended Data Fig. 1a). To perform segmentation on degraded images, one could first apply an image restoration method. Classical approaches for image restoration use various types of linear and nonlinear filtering[11–13], sometimes in combination with matrix decomposition methods[4,14,15]. Modern approaches instead use deep neural networks, but this requires a training dataset of paired clean and degraded images[16,17]. Neural networks for deblurring

or upsampling may sometimes require additional, image-specific knowledge such as the point-spread function[16–22].

The requirement for pairs of clean and corrupted images can be difficult to satisfy in practical experimental settings. To remove this requirement, methods like Noise2Self and Noise2Void learn to predict denoised pixel values from the context around each pixel[23,24]. These methods do not require clean images, but rely on the independence of noise across pixels and cannot be extended to other types of image restoration. Another possible limitation of existing denoising techniques is their reliance on a pixel reconstruction loss, which may be suboptimal for subsequent image analysis tasks like segmentation. A denoising loss has also been used jointly with a segmentation loss to improve segmentation especially in the limit of scarce training data[25].

In contrast to these approaches, here we asked whether images can be restored specifically for the purpose of improved segmentation and whether a single model trained on a varied dataset can generalize out of the box to new images. We answer the first question by chaining together two neural networks: a trainable restoration network followed by a pretrained Cellpose segmentation network. We train the former network to minimize the cost function of the latter one. In addition, we employ perceptual losses to make the reconstruction perceptually similar to the clean images[26,27] and we use a balanced combination of large-scale datasets for training[28–34]. We start below by describing and

HHMI Janelia Research Campus, Ashburn, VA, USA. ✉e-mail: stringerc@janelia.hhmi.org; pachitarium@janelia.hhmi.org

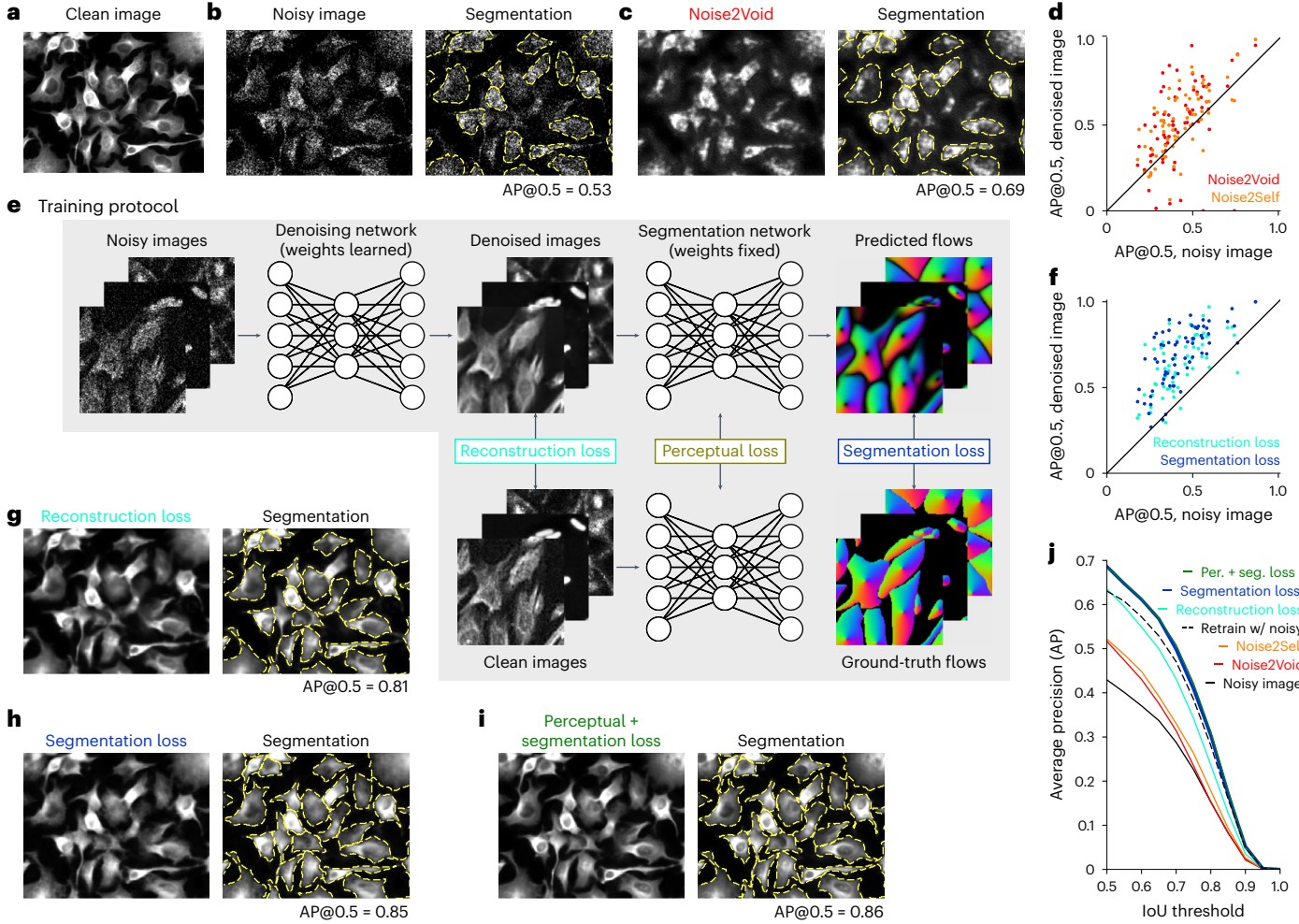

**Fig. 1 | Generalist denoising model. a**, Original (clean) image from the Cellpose test set. **b**, Same image with added Poisson noise (left) and segmentation (right). AP at an IoU threshold of 0.5 (AP@0.5) reported. **c**, Noisy image in **b** denoised with Noise2Void (left) and then segmented (right). **d**, Improvement in segmentation performance after denoising with Noise2Void or Noise2Self for 68 test images. **e**, Training protocol for the Cellpose3 denoising network. **f**, Same as

**d**, for denoising with the reconstruction loss network (teal) or the segmentation loss network. **g–i**, Denoised images (left) and segmentations (right) for networks trained with different losses. **j**, Mean AP score as a function of IoU threshold across 68 test images, for the noisy and denoised images; dashed line shows a model directly trained to segment noisy images. IoU, intersection over union.

demonstrating our approach in the denoising case, then we proceed to extend the approach to deblurring and upsampling.

## Results

### Model design and validation

Many imaging methods are affected by per-pixel noise[4,5] that can substantially reduce the performance of segmentation algorithms, such as Cellpose (Fig. 1a,b). This may be due to sensor noise, which is often described as Gaussian, or shot noise, which is well captured by a Poisson distribution. Both types of noise can be avoided with higher stimulation intensities or longer exposures, but this often results in degradation of the sample or long imaging times that may be unacceptable. To create a diverse dataset of images with pixel noise, we added different amounts of Poisson noise to images of cells and other objects from the Cellpose test set[28]. As Poisson noise becomes Gaussian-like in the limit of high simulated brightness, this approach should model both the sensor and the shot noise. To create a sensitive benchmark, we added sufficient noise to degrade the segmentation performance by a factor of ~2.

First, we applied two established techniques to the test data: Noise2Void and Noise2Self[23,24]. These methods can blindly denoise images without requiring clean samples, which makes them directly applicable to our denoising test set, although they require training a

new deep-learning model in each case, which we did here (Fig. 1c,d). When assessing the segmentation quality using the average precision (AP) metric (Methods), we find that the denoised images indeed had higher segmentation accuracy compared to the noisy images, but the accuracy also decreased for a subset of the images (Fig. 1d).

To improve on these approaches and remove the requirement of training at test time, our method learns a single denoising network for the entire Cellpose training set, which consists of diverse cellular and noncellular images from various sources (Fig. 1e and ref. 28). During training, Poisson noise of randomly varied magnitude was added to the images on each training batch. We explored three different training objectives: the standard pixel reconstruction loss[16,23,24], a novel segmentation loss and a perceptual loss[26,27]. Training with the pixel reconstruction loss was effective at restoring image quality on a variety of test images, but sometimes removed or blurred the boundaries between cells, thus preventing successful segmentation (Fig. 1g,j).

To improve on this, we directly trained the denoising network to output images that segment well, rather than images which reconstruct well (Fig. 1e). We achieved this with a segmentation loss, computed by running the denoised images through the Cellpose 'cyto2' segmentation network[28] and comparing the outputs to the ground-truth segmentations. The segmentation network remains

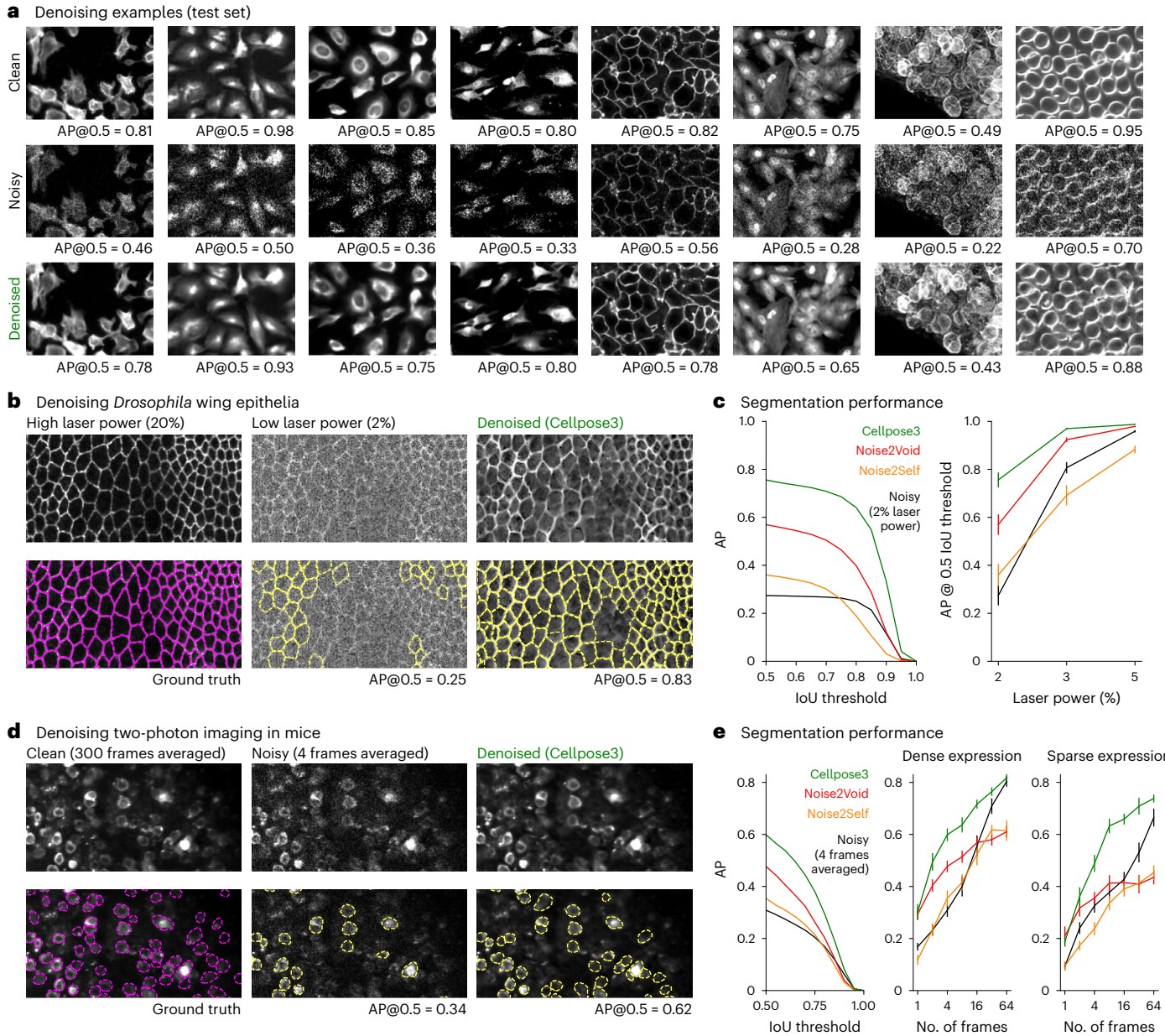

**Fig. 2 | Denoising cells. a**, Ground-truth images from the Cellpose test set (top), with Poisson noise added (middle) and denoised with Cellpose3. AP@0.5 reported. **b**, Images of *Drosophila* wing epithelial cells[16] acquired with high laser power (left), low laser power (middle) and denoised with the Cellpose3 network (right) (top). Segmentations (bottom). The high laser power segmentation was used as ground truth. **c**, Mean AP score across 26 test images either original or denoised with various methods (left). AP score at IoU threshold of 0.5 across laser powers (right). Error bars represent s.e.m., *n* = 26 test images. **d**, Same as **b**, for in vivo imaging of cortical neurons averaged across 300 frames (left), across four frames (middle) and then denoised (right). **e**, Mean AP score across 12 test images either original or denoised with various methods (left). AP score at IoU threshold of 0.5 as a function of the number of averaged frames (middle and right). Error bars represent s.e.m., *n* = 12 test images (middle) and *n* = 8 test images (right).

fixed during training and only the denoising network changes. This ensures that the denoising network outputs images that 'look right' to the pretrained segmentation network. The training error thus back-propagates through the segmentation network, without changing it and then further through the denoising network, producing gradients for learning. Training with the segmentation loss resulted in images where cell boundaries were more clearly distinguishable (Fig. 1h,j); however, the restored images were still perceptually distinguishable from the clean images. While not detrimental for segmentation, the appearance may be distracting for human observers, especially when training new models with the human-in-the-loop approach from Cellpose2 (ref. 35).

To maintain overall visual fidelity to the ground-truth images, we aimed to reconstruct abstract features of the images, rather than pixel values directly. This can be achieved with perceptual losses, which can be used to reconstruct the covariance matrix of feature activations in a pretrained neural network[26,27]. Rather than using a standard ImageNet-pretrained model, we used the Cellpose model itself, which contains abstract information about cell appearance at different levels of its hierarchy (Fig. 1e). Combining the perceptual and segmentation losses results in our final denoising network, which we will refer to as Cellpose3 (Fig. 1i,j). As a further comparison, we retrained the original Cellpose model directly on the noisy images, to output segmentations without an intermediate denoising step. This approach did not

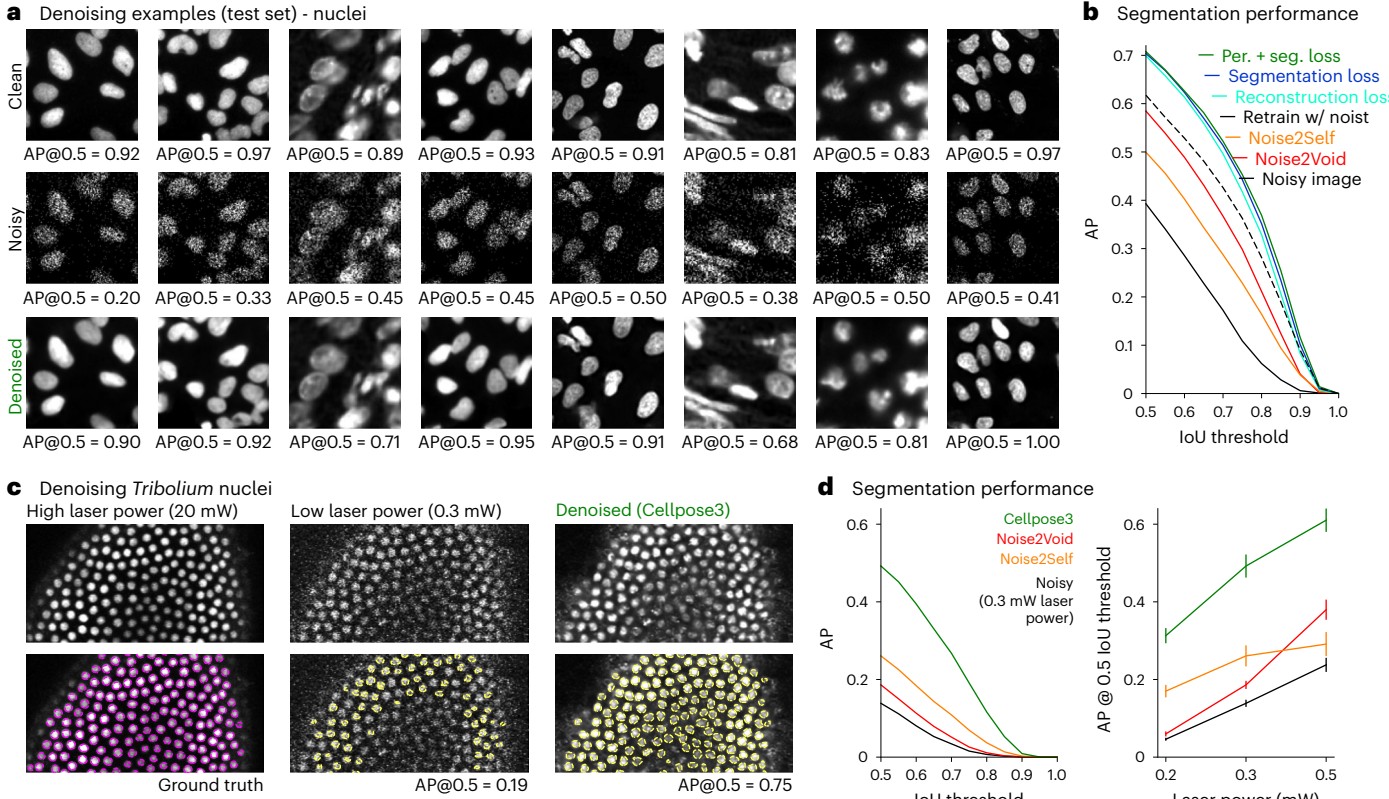

**Fig. 3 | Denoising nuclei. a**, Example ground-truth image crops from the nuclei test set (top), with Poisson noise added (middle) and denoised (bottom). AP@0.5 reported for entire, uncropped image. **b**, Mean AP score averaged across 111 test images for noisy and denoised images; dashed line shows a model directly trained to segment noisy images. **c**, Images of *Tribolium* nuclei during development[16], acquired with high laser power (left), low laser power (middle) and denoised with Cellpose3 (right). Segmentations (bottom). The high laser power segmentation was used as ground truth. **d**, Mean AP score across six test images either original or denoised with various methods (left). Mean AP score at IoU threshold of 0.5 across laser powers (right). Error bars represent s.e.m., *n* = 6 test images.

perform as well as segmenting images that were denoised by Cellpose3 (Fig. 1j). This model benefited from the large size of the Cellpose training set because performance continued to increase as a function of the number of training images (Extended Data Fig. 1b). We also found that Cellpose3 denoising increased the segmentation quality of other segmentation algorithms such as Stardist (Extended Data Fig. 1d)[36].

The Cellpose3-denoising network successfully reconstructed a large variety of test images, not seen during training, recovering both the fine cell boundary details as well as the visual appearance of different types of microscopy data (Fig. 2a and Extended Data Fig. 2a). One concern with all image reconstruction methods is that they could hallucinate cells on test images[37]. Our method allows for testing this possibility quantitatively by investigating whether false-positive cells are detected on the reconstructed images. We found that the false-positive rates were similar across all denoising approaches and similar to the false-positive rate of the original noisy images (Extended Data Fig. 2b–d). Thus, we do not find quantitative evidence for hallucinations.

Next we benchmarked Cellpose3 on data with shot noise acquired experimentally from ref. 16 and collected in our laboratory. We used the *Drosophila* wing epithelial cell dataset from ref. 16, which consisted of fields of view acquired at different laser powers using a spinning disk confocal microscope. To perform this benchmark, we first obtained segmentations of the images acquired at very high laser power using the original Cellpose1, which were near perfect by visual inspection (Fig. 2b). We used these segmentations as ground truth for evaluating the reconstructions of images acquired at low laser powers. We find that Cellpose3 substantially increases the segmentation quality of these images and more so than previous methods, across a range of laser powers (Fig. 2b,c). To generate another ground-truth dataset,

we used two-photon imaging to record neurons in mouse visual cortex expressing jGCaMP8s for 300 time points at 30 Hz (ref. 38). We averaged time points to create more or less noisy data, using the data averaged over all frames as the clean sample (Fig. 2d). Application of Cellpose3 improved segmentation performance in this dataset as well, in both dense expression areas and sparse expression areas (Fig. 2d,e).

Finally, we wanted to ask how well Cellpose3 performs in cases where many images of the same type are available at test time. In such cases, retraining the blind denoising methods (Noise2Self and Noise2Void) may perform better compared to training on a single test image, which we confirmed on the Cell Image Library dataset CCDB:6843 consisting of 100 similar images[39] (Extended Data Fig. 3a). Nonetheless, the Cellpose3 approach still outperformed these previous approaches both in visual quality and segmentation quality (Extended Data Fig. 3b,c). In addition, we trained CARE (content-aware image restoration; Methods) using pairs of noisy and clean training images from the training data of the specialized dataset[16] (Methods). Cellpose3 also outperformed this approach (Extended Data Fig. 3b,c).

We next applied the Cellpose3 approach to a large 'Nuclei' dataset of nuclear images from various sources[31,40–42], which were previously used to train the Cellpose 'nuclei' segmentation model[28] (Fig. 3a and Extended Data Fig. 4a). In this case, we found that all three losses (reconstruction, perceptual and segmentation) produced almost identical segmentation performance on the test set. We suspect this is because nuclear segmentation relies on simpler object shapes which may be easier to denoise. Nonetheless, the Cellpose3 approach outperformed the blind denoising methods as well as the segmentation model trained directly on the noisy images (Fig. 3b). Similar to the cellular denoisers, the nuclear denoisers did not hallucinate nuclei, as quantified by the

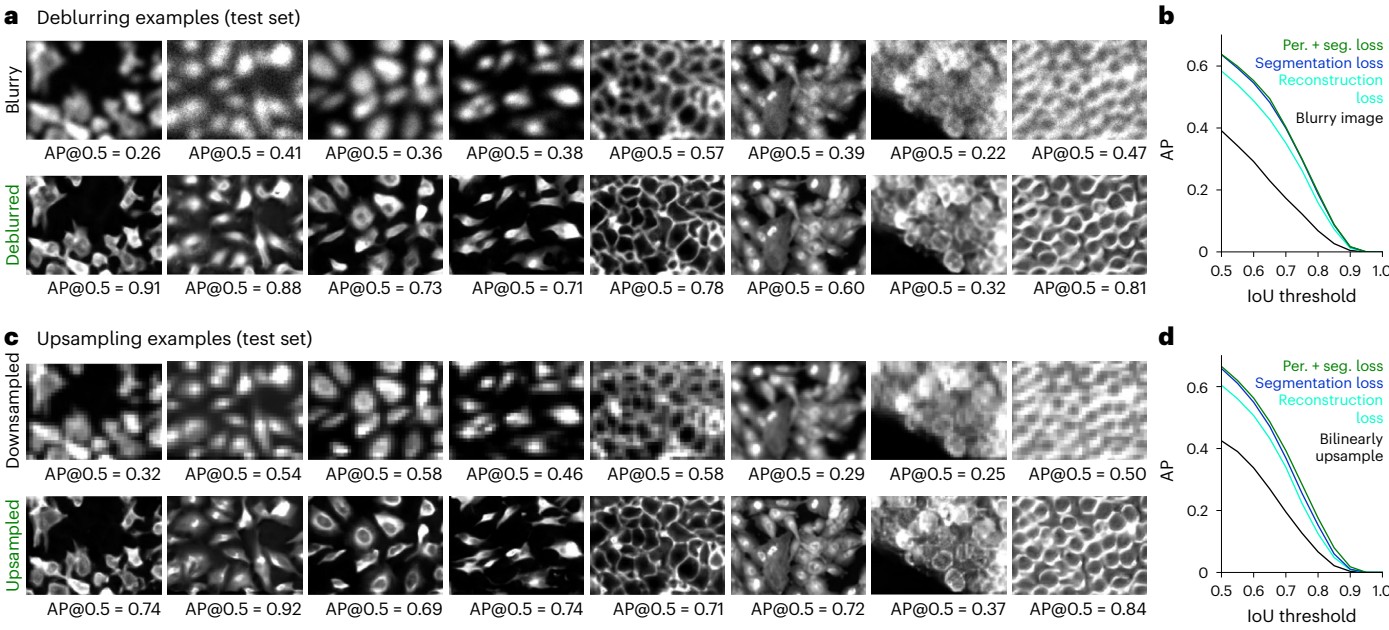

**Fig. 4 | Deblurring and upsampling cellular images. a**, Same clean image crops as shown in Fig. 2a, with Gaussian blurring (top) and deblurred with Cellpose3 (bottom). AP@0.5 is reported for entire image. **b**, Mean AP score across 68 test images either blurry or deblurred. **c,d**, Same as **a,b** for downsampled images that were upsampled with a Cellpose3 network.

false-positive, true-positive and false-negative rates (Extended Data Fig. 4b–d). On images of *Tribolium* nuclei acquired at different laser powers using a laser scanning confocal microscope from ref. 16, the Cellpose3-denoised images resulted in segmentations, which matched the clean image segmentations substantially better than the other denoising approaches (Fig. 3c,d).

### Deblurring and upsampling

Next we considered whether the same approach can be applied to other image restoration tasks, such as deblurring and upsampling. Blurring in microscopy is typically due to scattering such as in thick tissue imaging. Computational deblurring methods (sometimes referred to as 'deconvolution') can reveal finer details than are visible in the blurred images. Similarly, undersampling of a field of view is also common, often due to speed or photon-budget constraints. Upsampling such images (sometimes referred to as 'super-resolution') can restore details that make the segmentation better.

For deblurring, we trained our networks on images with Gaussian blur of random spatial sizes. For test images, we again fine-tuned the amount of degradation per image to approximately decrease the segmentation performance by half. Networks trained on the Cellpose training set were able to deblur a variety of cellular images at test time (Fig. 4a), improving the segmentation performance substantially (Fig. 4b). We also trained the networks on the nuclei training set and found a large improvement in segmentation performance after deblurring (Extended Data Fig. 5a,b). The networks trained with both the perceptual and segmentation loss again performed best (Fig. 4b and Extended Data Fig. 5b). This method also worked well in cases where only part of an image was blurred (Extended Data Fig. 6). We note that more elaborate 'blurring' functions may be needed for complex applications such as imaging in deep tissue and these may require training a new model[9,17].

For upsampling, we trained networks on images that were downsampled by decimation. The test set was again constructed to approximately reduce segmentation performance in half. The upsampling networks trained on the Cellpose or nuclei datasets recovered some of the visual detail in the test images and improved segmentation

(Fig. 4c,d and Extended Data Fig. 5c,d). As before, networks trained with the segmentation loss performed best.

In most microscopy methods, the point-spread function and sampling frequency are larger in the axial ($z$) dimension than in the $xy$ plane. In thick tissue, this results in blurry and downsampled data in the axial dimension. We thus tested whether the Cellpose3 approach could reduce image degradation in the axial slices of a three-dimensional (3D) volume, the $zy$ and $zx$ slices. We trained Cellpose3 networks for cells and nuclei to correct axial degradations and applied them to a 3D stack with ground-truth segmentations[28]. Indeed, axial restoration with Cellpose3 greatly improved 3D segmentation performance (Extended Data Fig. 7). We note that this two-dimensional (2D)-based approach does result in slight discrepancies of the results on consecutive slices and may sometimes require smoothing or a 3D specialist restoration method to reconcile differences and integrate results into a 3D segmentation.

In summary, the Cellpose3 networks for denoising, deblurring and upsampling improved segmentation performance for a variety of imaging modalities (Extended Data Fig. 8).

### One-click image restoration

So far we have shown separate models trained on cells and nuclei, separately for denoising, deblurring and upsampling. There are other datasets that we could have used such as TissueNet[30], LiveCell[29], Omnipose[32], Yeaz[34] and DeepBacs[33]. Instead of training new restoration models in each case, we asked whether single restoration models could be trained on all these datasets combined and across degradation types.

Training a single restoration model across all datasets required first training a super-generalist 'cyto3' segmentation network, to be used for computing the segmentation and perceptual losses (Fig. 5). Similar to the Cellpose2 paper[35], we observed a small but consistent loss in performance from the network trained on all datasets (for example, AP@0.5 on the Cellpose test set was 0.786 versus 0.771 for the dataset-specific versus the super-generalist 'cyto3' model). This result differs markedly from a recent study[43], which found a much larger decrease in performance for generalist versus specialist Cellpose models, likely due to the suboptimal training regimen used there (see Methods for more details). Here we adjusted the sampling

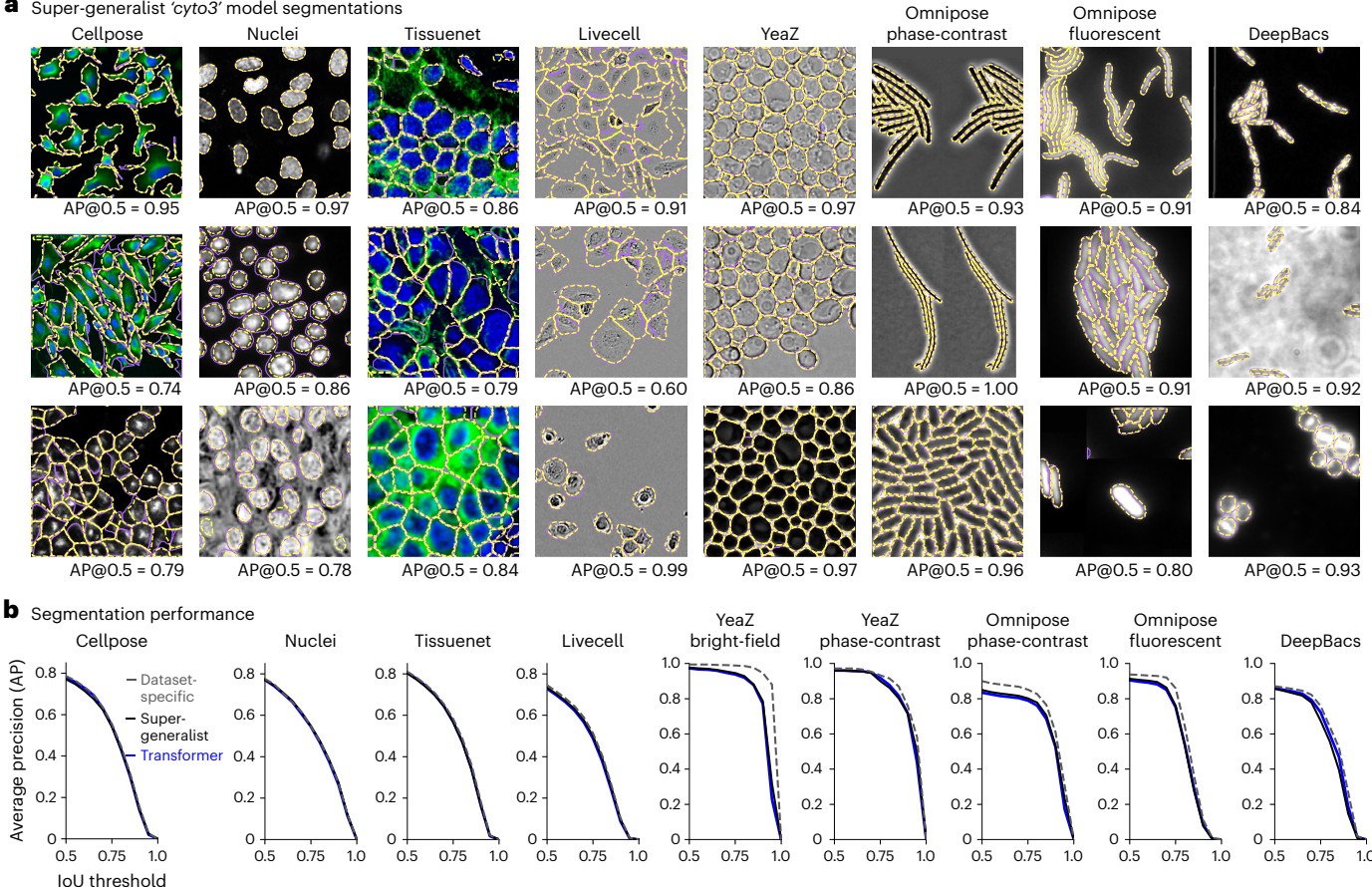

**Fig. 5 | Segmentation performance of the 'super-generalist' model on clean images.** A single model was trained to segment nine datasets of cellular and nuclear images. **a**, Three example images from each test set (YeaZ bright-field and phase-contrast are combined), with ground-truth segmentations overlaid in purple and super-generalist model segmentations in yellow. The AP@0.5 is the AP of the segmentation at an IoU threshold of 0.5. **b**, Segmentation performance on the different test sets, averaged across all test images, of the super-generalist model, the dataset-specific models and the transformer model trained on all datasets.

probability of the datasets during training, to prevent large but homogeneous datasets like TissueNet and LiveCell from dominating the statistics of the cost function. We also tested a transformer backbone for Cellpose and found that it did not improve performance[44,45]; in fact, it had nearly identical performance to the standard residual U-net backbone, which suggests that generalization performance is limited by dataset diversity rather than model architecture. Further demonstrating the importance of the training dataset, we found that out-of-distribution generalization was very limited for all algorithms we tested (Extended Data Fig. 9).

We first trained separate networks for denoising, deblurring and upsampling of all the training images from the combined nine datasets ('degradation-specific' models). This resulted in networks that performed as well as the dataset- and degradation-specific models in all cases, except for deblurring and upsampling of nuclei (Fig. 6a–c). We also trained models across all degradation types but separately for the Cellpose dataset and the nuclei dataset ('dataset-specific' models), achieving close to optimal performance. Finally, we trained a model on all degradation types and all datasets. We refer to this as a 'one-click' model, as it can be run without specifying the degradation or dataset type. This model performed similarly to the degradation-specific models on the Cellpose and nuclei test set. The model also worked well on the other seven datasets (Fig. 6d and Extended Data Fig. 10). We provide this model in the graphical user interface (GUI) along with the dataset-specific and degradation-specific models, which should be useful especially for deblurring and upsampling.

## Discussion

Here we introduced a new method for restoring biological images, which optimizes a downstream task such as segmentation, rather than optimizing image reconstruction. Like the blind deconvolution methods[23,24], our approach does not require clean images at test time. By training this method on a large and diverse dataset, we achieved high out-of-sample generalization on new images.

Although our method focuses on segmentation rather than reconstruction, the restored images are nonetheless visually compelling due to the use of a perceptual loss. Note that the perceptual loss does not improve segmentation performance; we nonetheless include it because it improves the appearance of images and this may help users create manual annotations with the human-in-the-loop approach from Cellpose2. We do not report here some of the standard denoising metrics like PSNR, as these metrics are essentially a way to estimate the per-pixel reconstruction loss. Our approach is not designed to reproduce exact pixel values, which may in general be impossible from limited information, but rather to reproduce more abstract features, such as cell boundaries and their perceptual appearance.

It is also important to discuss the use of denoised images in scientific research. As modern denoisers use complex image priors learned from their training datasets, there is always a risk that certain image features are 'invented' by the network rather than truly being present in the data. With respect to Cellpose3, one could take two approaches: (1) a conservative approach, where Cellpose3 is used to detect and segment cells, but the biological quantifications are conducted on

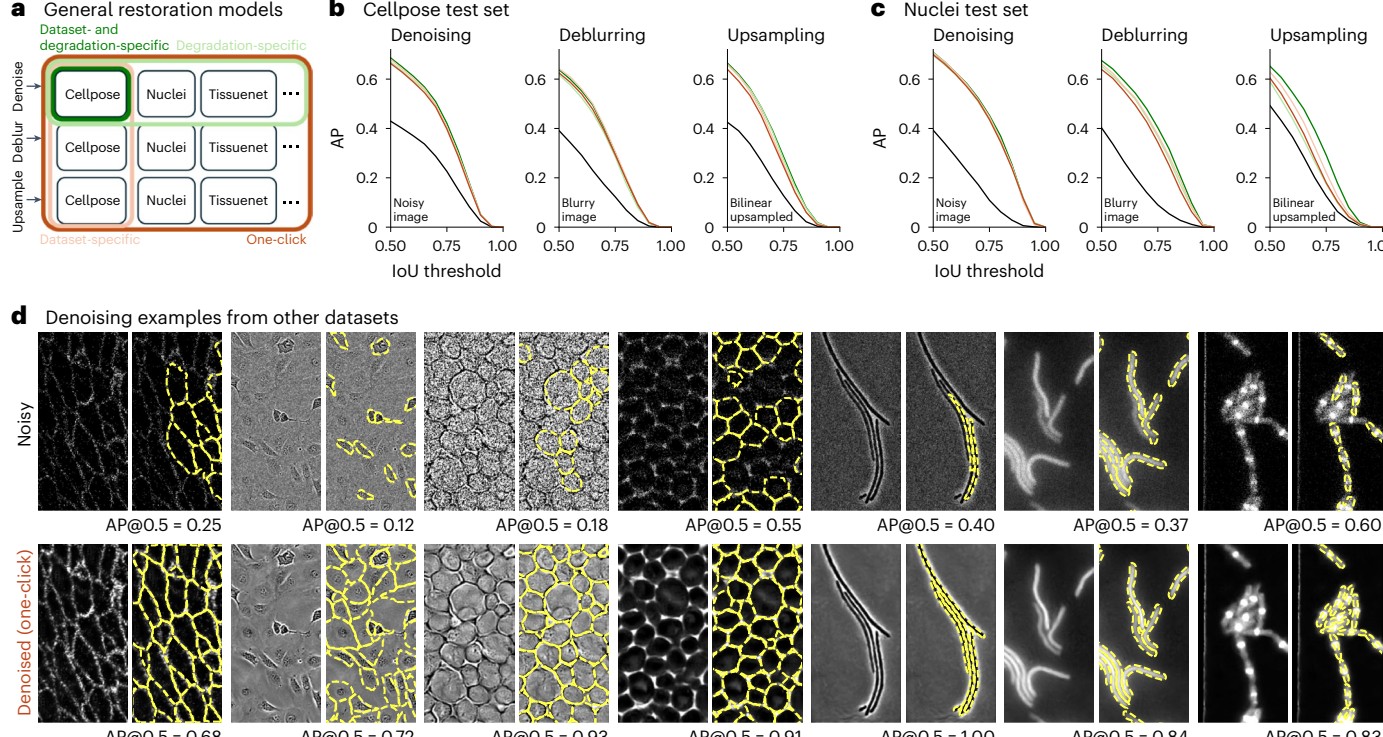

**Fig. 6 | One-click model for denoising, deblurring and upsampling.**
**a**, Schematic of training datasets and degradation types for different networks.
**b**, Segmentation performance on the Cellpose test set using the 'cyto2' segmentation model on images after denoising (left), deblurring (middle) or upsampling (right) with the one-click model compared to the dataset-specific, degradation-specific and dataset- and degradation-specific models. **c**, Same as **b** for the nuclei test set and 'nuclei' segmentation model. **d**, Example images and their segmentations from other datasets with Poisson noise added (top) and denoised with the one-click model (bottom). AP@0.5 reported.

the raw images; or (2) a more liberal approach, where all analyses are carried out on the denoised images. We believe the more-conservative approach should be used whenever possible. When this is not possible, the liberal approach could be used if well-matched control images are available to verify the method. For example, pairs of high/low-quality images could be obtained for a small subset of the data to serve as an internal control.

## Online content

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

## Methods

The Cellpose code library is implemented in Python 3 (ref. 46), using pytorch, numpy, scipy, numba, opencv, imagecodecs, tifffile, fastremap and tqdm[47–55]. The GUI additionally uses PyQt, pyqtgraph and superqt[56–58]. The figures were made using Matplotlib and Jupyter notebook[59,60]. All experimental procedures were conducted according to Institutional Animal Care and Use Committee (IACUC), ethics approval received from the IACUC board at Howard Hughes Medical Institute Janelia Research Campus.

### Cellpose3-denoising network training and testing

**Model architecture.** We used the same model architecture as the Cellpose model, described in detail elsewhere[28]. The Cellpose model is a deep neural network with a U-net based architecture, consisting of four downsampling blocks and four upsampling blocks, each block containing four convolutional layers with residual connections[61,62]. The model takes as input a single noisy image and outputs the image denoised. On 224 × 224 images, the network runs at a speed of 750 images per second on a server A100 GPU, or 210 images per second on a consumer-grade A4000 GPU.

**Training.** All training was performed with the AdamW optimizer[63]. The learning rate increased linearly from 0 to 0.001 over the first 10 epochs, then decreased by factors of 2 every 10 epochs over the last 100 epochs. Each network was trained for 2,000 epochs. We also trained denoising networks with varied numbers of epochs and random subsets of images to show scaling properties (Extended Data Fig. 1b,c). The one-click models were initialized using the weights from the Cellpose3 networks trained with the 'cyto2' dataset; all other models were trained from scratch.

The reconstruction loss was the mean squared error between the ground-truth image and the predicted image (the network output). To compute the segmentation loss, we used the 'cyto2' model, the 'nuclei' model, or the new super-generalist model. The segmentation loss was the same loss as the Cellpose segmentation loss[28]: the mean squared error between the $xy$ flows from the ground-truth segmentation and the predicted $xy$ flows, scaled by a factor of five, added to the binary cross-entropy between the ground-truth cell probability and the predicted cell probability. For the perceptual loss we computed the correlation matrix of neural network activations at each downsampling block[26,27]. A target correlation matrix was computed for the ground-truth image and a predicted correlation matrix was computed for the denoised/deblurred/upsampled image. The perceptual loss function was defined as the mean squared error between these two matrices, normalized by the s.d. of the target correlation matrix in each downsampling block. This latter normalization ensures that each block level contributes approximately the same variance to the cost function.

All images were normalized such that 0 was set to the first percentile of the image intensity and 1 was the 99th percentile of the image intensity. The image intensities were then clipped at 0, which was necessary to add Poisson noise.

During training, all images were resized such that the cell or nuclei diameters were 30.0 or 17.0 pixels, respectively. Then, before adding synthetic noise, the images were resized 0.5–2 times their original size and randomly cropped to a size of 340 × 340. This allowed the Poisson noise to be added at various scales. After adding the noise, blurring and/or downsampling, the images were randomly resized and cropped to 224 × 224 using the original Cellpose augmentations: random rotation, random flipping and random resize with a scale factor between 0.75 and 1.25, such that 1.0 is equivalent to cell or nuclei diameters of 30.0 or 17.0 pixels. When training the one-click models using the super-generalist segmentation model, all images were resized such that the cells or nuclei diameters were 30.0 pixels. When training the one-click models, the images across all datasets were sampled during each epoch, as described in the super-generalist training section.

**Synthetic noise.** We added three types of degradations to images: Poisson noise, Gaussian blurring and downsampling. During training, for the Poisson noise generation, we multiplied each image by a random scaling factor, drawn from a gamma distribution with $\alpha = 4.0$ and varying values of $\beta$ and then used this scaled image to draw a random sample of Poisson noise. Poisson noise was generated on 80% of all training images, with $\beta = 0.7$ during denoising training, $\beta = 0.1$ during deblurring training and $\beta = 0.03$ during upsampling training. This operation was applied last.

For the deblurring and upsampling training, we blurred the images 80% of the time with an isotropic 2D Gaussian filter with an s.d. drawn from a uniform distribution between 1 and 10. These s.d. values were then scaled by the ratio between the diameter of the cells/nuclei in the image divided by 30.

For the upsampling training, we downsampled the images 80% of the time, with a downsampling factor drawn as a random integer between 2 and 7. The s.d. of the Gaussian blurring for these images was equal to the downsampling factor multiplied by ~0.4. The images were first blurred, then downsampled using subsampling with the random integer, then bilinearly interpolated to the original size to be input into the network. After this step, the Poisson noise was added.

For training with all degradation types, each image was randomly degraded with the denoising, deblurring or upsampling protocols above, with a probability of one-third for each.

For the anisotropic network training, we used $\beta = 0.1$ for the Poisson noise and blurred the images 80% of the time, with an s.d. drawn from a uniform distribution between 1 and 18. The s.d. of the Gaussian in the $x$ direction was ten times smaller. We also downsampled the images 80% of the time in the $z$ direction with a downsampling factor drawn as a random integer between 2 and 12 and for these images the blurring was equal to the downsampling factor multiplied by ~0.5. Random rotation augmentations were turned off to ensure the $z$ direction (with downsampling) was consistent across training samples.

For testing, we added different amounts of noise/blurring/downsampling to each image, scaled to reduce the segmentation performance of 'cyto2' or 'nuclei' on the image by 50%, as measured by the AP at an IoU threshold of 0.5. The scaling factors for Poisson noise varied from 0.5 to 80.0, the Gaussian blurring s.d. values varied from 0.5 to 10.0 and the downsampling factors varied from 2 to 6. When generating the blurry images, a small amount of Poisson noise was added, with a scale factor of 120.0. For images with partial blurring, we blurred part of the image with 1.25 times the Gaussian s.d. from the fully blurred images. When generating the downsampled images, blurring was performed with an s.d. proportional to half the downsampling factor. For the anisotropic downsampling, the downsampling was only performed axially in the 3D dataset and the Gaussian blurring s.d. was equal to the downsampling factor in the axial direction and ten times smaller in the nonaxial direction.

After these operations, each of the noisy images was normalized such that 0 was the first percentile and 1 was the 99th percentile.

**Images with real noise.** We used two imaging datasets from elsewhere[16] collected at varying laser powers to demonstrate the ability of Cellpose3 to denoise images with real Poisson shot noise (Figs. 2b,c and 3c,d)[64]. The *Drosophila* wing epithelia cell dataset consisted of 26 images collected at four different laser powers. Each image consisted of multiple planes; we used the maximum projection image provided in the paper. The *Tribolium* nuclei dataset consisted of six images collected at four different laser powers. Each image consisted of multiple planes; we used the maximum projection image across 16 planes in the middle of the volume. Ground-truth segmentations for the images were not provided, so we used the Cellpose segmentations of the high laser power images as the ground truth. Based on the average region of interest size in the high laser power segmentation, we set the diameters to 24 and 12 for the epithelia and nuclei datasets, respectively

and used these values for all laser powers and for the denoised images. For the *Drosophila* wing epithelia cells (Fig. 2b,c), we set the cell probability threshold to −2.0 for all segmentation evaluations to ensure the masks filled the cytoplasmic space across noise levels and the denoised images.

We also collected a new dataset of noisy images using in vivo two-photon imaging in mice expressing somatic jGCaMP8s (riboL1-jGCaMP8s mouse line)[38,65] as described elsewhere[66]. We imaged each neural field of view for 300 time frames at a frame rate of 30 Hz. Twelve fields of view had dense expression and eight had sparse expression. As ground truth for the segmentations, we used the average of 300 frames. Based on the average region of interest size in the time-averaged image segmentation, we set the diameter to 17 for use for all denoising and segmentation.

## Segmentation networks

**Cellpose model.** The Cellpose model architecture is described in detail elsewhere[28]. The Cellpose model predicts three outputs: (1) the probability of a pixel being inside a cell; and the flows of pixels toward the center of a cell in (2) $x$ and (3) $y$. The flows are then used to construct the cell masks. The Cellpose model, 'cyto2', was trained on 796 images of cells and objects with one or two channels (if the image had a nuclear channel), with an average cell diameter of 30.0 pixels used in training. The Cellpose model, 'nuclei', was trained on 1,025 single-channel images of nuclei, with an average nucleus diameter of 17.0 pixels used in training.

The Cellpose model can be extended to 3D by computing the flows on $xy$, $xz$ and $yz$ slices and then averaging the flows across these views[28]. We used this approach to segment a 3D volume from mouse cortex (described elsewhere[28]). We manually annotated this volume, resulting in ground-truth 3D segmentations, which we used for segmentation benchmarking with and without anisotropic deblurring/upsampling.

**Noisy image training.** The 'retrain w/ noisy' networks were Cellpose networks trained from scratch in the same way as the Cellpose 1.0 paper: 500 training epochs with a batch size of 8, stochastic gradient descent optimizer, weight decay of 0.00001, momentum of 0.9 and learning rate of 0.2. The learning rate increased linearly from 0 to 0.2 over the first 10 epochs, then decreased by factors of 2 every 10 epochs after the 400th epoch. The proportion and scaling of noise added to the images was the same as that added to the Cellpose3 networks trained on the same noise type.

**Dataset-specific, super-generalist and transformer training.** The dataset-specific and super-generalist models use the same architecture as the original Cellpose model which we found to be sufficient. Increasing the width of the layers or the depth of the network did not help (results not shown). Recent analyses suggest that super-generalist models require transformer models and perform better than Cellpose[43,67], but we believe the comparisons have not been carried out correctly. For example, one analysis[67], allows the transformer model to retrain but does not retrain Cellpose. As another example, other analyses[43,68] trained Cellpose using the finetuning protocol from the Cellpose2 paper, rather than the protocol for training Cellpose from scratch and the training dataset was dominated by TissueNet images, which resulted in class imbalance, a known problem for training supervised models. It may be that transformers are less affected by such training set imbalance, but more likely the very long training times of transformers compensate for the class imbalance. Even transformers can benefit from class balancing, as the winner of the NeurIPS 2022 cell segmentation challenge (MEDIAR[44]) used a complex balancing strategy.

To make the comparison to transformers more direct, we replaced only the backbone of Cellpose with a transformer, keeping everything else the same. This is similar to the approach of the MEDIAR algorithm. This transformer consisted of a transformer-based segformer encoder

and a multi-scale attention decoder, which was trained to predict the cell flows and probabilities from Cellpose[69,70]. As in the MEDIAR paper, we used the implementation of the encoder and decoder from the segmentation_models.pytorch GitHub[45]. Also, as in the MEDIAR paper, we used the MIT-B5 segformer encoder and initialized the encoder with weights from pretraining on imagenet provided in the GitHub repository.

To optimize these models, we used the AdamW optimizer[63]. The learning rate increased linearly from 0 to 0.005 over the first 10 epochs, then decreased by factors of 2 every 10 epochs over the last 100 epochs. The super-generalist and transformer models were trained for 5,000 epochs and the dataset-specific model was trained for 2,000 epochs. For the super-generalist and transformer models, each epoch consisted of 800 training images, randomly sampled from the total training set size of 8,402 images. For the dataset-specific models, each epoch consisted of all training images in the dataset, if there were fewer than 800 training images, or consisted of 800 images randomly sampled from the dataset. Each image had one or two channels, where the second channel contained the nuclear channel if it was a cellular image with two channels, or otherwise was set to zero. The nuclei-only images had the nuclear channel as the first channel and the second channel set to zero. We used the original Cellpose augmentations, as described in the denoising section, with an average cell/nuclei diameter of 30.0 pixels for all images.

For the super-generalist and transformer models, the sampling probability of each image varied depending on the image set: PhC yeast images and fluorescent bacterial images were sampled at a probability of 1% each; bright-field yeast images, phase bacterial images and Deep-Bacs images at 2% each; LiveCell images at 5%; TissueNet images at 8%; nuclei images at 20%; and cyto2 images at 59%. We upweighted images in the cyto2 and nuclei training sets because they contained the most variability across images. These sampling probabilities and 800 images per epochs were also used when training the super-generalist size model and also for the one-click model training. For the size network training, we used the same defaults as in the Cellpose paper: ten epochs of training images and an L2 regularization constant of 1.0 (ref. 28).

The flow representations of the masks were modified slightly from the original method[28]. We used the point within the mask closest to the center of mass as the source of the diffusion process, instead of the median. In cells with a long process, like a dendrite, for example, this puts the source closer to the process rather than in the soma, to improve convergence of such longer processes. We also no longer logarithmically scaled the diffusion probabilities before computing the gradients and when normalizing the gradients by their L2 norm, we added a smaller epsilon to the L2 norm ($1 \times 10^{-60}$ versus $1 \times 10^{-20}$). This increased the size of smaller gradients, improving the performance on long and thin cells.

**Other segmentation models.** We compared the segmentation performance of Cellpose to other segmentation algorithms.

Mesmer uses a feature pyramid network with a ResNet50 backbone[62] to predict cell interiors, cell boundaries, background pixels and the distance of each cell pixel to the center of the cell[30]. We used the default pretrained model from DeepCell v.0.11.0. For non-TissueNet images, we rescaled each image so that the minimum pixel value was zero and the maximum pixel value was one, as the TissueNet images are scaled.

StarDist uses a U-net architecture to predict the distance of specified rays to the cell boundary[36]. We trained the model in ref. 28 on the Cellpose 'cyto' training set. During training, the images were rescaled so that the 1st percentile of the image was set to zero and the 99th percentile of the image was set to one and we performed the same normalization for the images in this study before applying the model.

MEDIAR, as described above, was trained by the paper authors on the Cellpose training and test sets, the LiveCell training and test sets,

the DSB2018 training and test sets, the Omnipose training and test sets and the NeurIPS challenge data, which included images acquired with the TissueNet platform[44]. We used the ensemble model described in the paper, which averages the outputs of two models trained with different protocols for prediction. For images with only one channel, we represented the images as grayscale with three equal channels and for images with multiple channels we placed the nuclei in the blue channel and the cytoplasm in the green channel.

**Evaluation.** The Cellpose segmentation models are trained such that all cells and nuclei are approximately the same size in pixels across all images, by resizing each image such that the average cell (or nuclei) diameter is 30.0 (or 17.0). Therefore, the images in the test set must be resized before running the segmentation models. In all analyses besides Fig. 5, the test images were rescaled per image using the average diameter computed from the ground-truth masks, rather than using the size estimation models from Cellpose. This was carried out because the size estimation models were trained on images that were not noisy, blurry or downsampled and so would not perform correctly on those images and also because the objective of the analysis is to assess the segmentation; we assume most users know the diameters of their cells or nuclei in their images and can provide them for the denoising and segmentation algorithms.

In Fig. 5, we evaluated the segmentation performance of the super-generalist model versus other models. For this, we did not assume that the network knows the average diameter of the cells/nuclei in the image and therefore used the super-generalist model to create a size model to estimate the diameters. Using these estimated diameters, each image was resized to set the average diameter to approximately 30 pixels and then run through each of the different segmentation models.

The flow error threshold (quality control step) was set to 0.4 and the cell probability threshold was set to 0. In Fig. 5, for all models we turned on test-time augmentations during evaluation, as in refs. 28 and [44]. When segmenting the phase-contrast images from the Omnipose dataset, we set the number of iterations 'niter' for the dynamics post-processing to 2,000 for all images, to improve the convergence for long and thin cells.

**Quantification of segmentation quality.** As described in Cellpose 1 and 2, we quantified the predictions of the segmentation algorithms by matching each predicted mask to the ground-truth mask that is most similar, as defined by the IoU between the predicted and ground truth. The average precision (AP) metric for each image is defined using the true positives (TP; matches with IoU above a given threshold), false positives (FP; predicted masks without matches) and false negatives (FN; missed ground-truth masks):

$$AP = \frac{TP}{TP+FP+FN}.$$

The AP reported is averaged over the AP for each image in the test set. We computed the TP rate, FP rate and the FN rate by normalizing the TP, FP and FN values per image by the total number of ground-truth masks per image, then averaging across images.

**Denoising model comparisons**
We compared the performance of the Cellpose3 models to Noise2Void, Noise2Self, DenoiSeg and CARE[16,23–25]. In Figs. 1 and 3a,b, we trained Noise2Void and Noise2Self on a per-image basis, as described in their papers, creating 68 models for the 68 noisy cellular test images and 111 models for the 111 noisy nuclear test images. In Figs. 2b,c and 3c,d, we also trained Noise2Void and Noise2Self on a per-image basis on each of the three noise levels, creating 78 models for the 78 noisy *Drosophila* epithelia test images and 18 models for the 18 noisy *Tribolium* nuclei test images. Each of these images was normalized such that 0 was the first percentile and 1 was the 99th percentile.

For specialist training, in Extended Data Fig. 3, we trained and tested Noise2Void, Noise2Self, DenoiSeg and CARE on images from the Cell Image Library dataset CCDB:6843 (ref. 39), consisting of 89 images in the training set and 11 images in the test set. We added 20 instances of random noise to each training image, drawn from the same distribution as used in Cellpose3 training and normalized in the same way. We then created a validation set from these images using quarter crops: one-quarter of each image was used for validation and the other three-quarters for training, resulting in three training crops per image. In total this resulted in 5,340 noisy training images and 1,780 noisy validation images. Each training and validation image had a ground-truth pair, the original image without noise, which was used for training CARE. DenoiSeg has a segmentation loss in addition to the Noise2Void loss and thus was trained with the segmentation masks on training images. Noise2Void, Noise2Self and DenoiSeg were additionally trained on the noisy test images, which were also divided into quarters, with one-quarter in the validation set and the other three-quarters in the training set. At test time, for each of the networks we used the weights from the epoch with the lowest validation loss.

**Noise2Self model.** Noise2Self is a blind denoising algorithm that does not use clean images for training and can be trained on single images[23]. For the single noisy image training, we first tried the 'Single-Shot Denoising' notebook in the Noise2Self GitHub, using the default CNN model defined with eight layers and the default training parameters: Adam optimizer, learning rate of 0.01 and 500 epochs; however, we found that the average segmentation performance on the denoised images from this training was poor (average AP@0.5 of 0.435 for cellular images). Instead, we used the U-net model used for CellNet training, as described in the Noise2Self paper. We used a batch size of eight and the same augmentations as Cellpose: random rotation, cropping and resizing (scale between 0.75 to 1.25) and flipping, with random crops of size 128 × 128. We trained each single image model with the Adam optimizer with a learning rate of 0.0005 and for 100 epochs. At test time, the test images were padded with zeros to achieve image dimensions that were divisible by 16 so that they could be run through the U-net and then the padding was cropped out on the output. This resulted in an average AP@0.5 of 0.511.

For the specialist training, we used the U-net model and a batch size of 64, as used for the CellNet training. We used the same transformations as the per-image training. We performed a sweep over learning rates (0.00001–0.01) to determine the best learning rate on the validation set based on the validation loss (mean squared error). We found that a learning rate of 0.0005 produced the lowest validation loss. We also tried different numbers of epochs, but found that, although training for more epochs reduced the validation loss, the test set segmentation performance became worse with more epochs. Therefore, we trained for 50 epochs, the setting used for the CellNet training. As in the per-image training, the images were padded then cropped during test time to enable them to be run through the U-net.

**Noise2Void model.** Noise2Void is also a blind denoising algorithm that does not use clean images for training and can be trained on single images[24]. For the per-image training we used the parameters and model from the denoising2D_SEM training notebook. This default model was a U-net with a depth of 2 and a kernel size of 3. The training used a validation set, so we divided each noisy test image into quarters and used one-quarter for validation. As suggested in the notebook, we set the number of epochs to 100, train steps per epoch of 1, learning rate of 0.0004, batch size of 128 (or less depending on the number of patches) and a patch size to 64 × 64 (for some images it was required to be size 60 × 60). We used the default augmentations, as defined in the function 'generate_patches_from_list': flipping and rotating by 90 degrees and all possible crops of the image into 64 × 64 patches.

For the specialist training, we used the same U-net model, the same augmentations and the default batch size of 128. We used the same transformations as the per-image training and used 25 training steps per epoch. We performed a sweep over learning rates (0.00001–0.01) to determine the best learning rate on the validation set based on the validation loss (mean squared error). We found that a learning rate of 0.0004 (default) produced the lowest validation loss. As with Noise2Self, we also tried different numbers of epochs, but found that, although training for more epochs reduced the validation loss, the test set segmentation performance became worse with more epochs. Therefore, we trained for 100 epochs, the setting recommended in the notebook.

**DenoiSeg model.** DenoiSeg is a blind denoising algorithm that also uses segmentation labels when available[25], which we trained using the specialist cellular dataset with noise added and with the segmentations for the training images. We used the default model configuration: a U-net with a depth of 2, a kernel size of 3 and 4 output maps (one with the denoised image and three for segmentation). The training used a validation set, so we divided each noisy test image into quarters and used one-quarter for validation. We trained using the default batch size of 128 and used 400 steps per epoch. We performed a sweep over learning rates (0.00001–0.01) to determine the best learning rate on the validation set based on the validation loss (mean squared error). We found a learning rate of 0.001 produced the lowest validation loss. We trained for the default number of epochs (40), finding that more epochs did not reduce the validation loss. We used the default augmentations, as defined in the function 'generate_patches_from_list': flipping and rotating by 90 degrees and all possible crops of the image into 128 × 128 patches.

At test time, we predicted the denoised images from the test images. We ran the 'cyto2' Cellpose network to segment these images. Additionally, we used the segmentations that are output from the DenoiSeg network. We chose the probability threshold based on the AP on the test set (to maximize performance), which resulted in a threshold of 0.45.

**CARE model.** The CARE model is a denoising algorithm trained to restore noisy images using noisy–clean image pairs. We only performed specialist training for CARE as it cannot be trained per image on noisy images.

For specialist training, we used the default CARE U-net model with a depth of 2 and a kernel size of 3, provided in the denoising2D training example notebook. We used the default parameters: two 128 × 128 patches per training/validation image, a training batch size of 8 and the number of training steps per epoch of 400 (equivalent to the number of images per epoch as Noise2Void). We turned off image normalization in CARE, as it was similar to our normalization and when on, it resulted in a subset of images with pixel values over $1 \times 10^{20}$. We performed a sweep over learning rates ($1 \times 10^{-5}$ to $1 \times 10^{-2}$) to determine the optimal hyperparameters on the validation set based on the validation loss (mean squared error). We found that a learning rate of $1 \times 10^{-3}$ and 100 epochs produced the lowest validation loss and using these hyperparameters chose the network with the lowest validation loss across training epochs (as is default in the CARE code). We then ran the test images through the network's 'predict' function.

### Segmentation datasets

We used nine publicly available datasets for training the one-click restoration models and for training the super-generalist 'cyto3' model.

**Cellpose cyto2 dataset.** This dataset consists of 618 images from the 'cyto' dataset (540 training images and 68 testing images), described in detail elsewhere[28], from various sources (https://idr.openmicroscopy.org/)[39,71–73] and an additional 256 new training images. The new training images were found using Google image searches or were submitted by users of Cellpose. The dataset is available at https://www.cellpose.org/dataset.

**Cellpose nucleus dataset.** This dataset of nuclear images was described in detail elsewhere[28]. It consists of 1,025 training images and 111 test images from various sources[31,40–42].

**TissueNet.** The TissueNet dataset consists of 2,601 training and 1,249 test images of six different tissue types collected using fluorescent microscopy on six different platforms (https://datasets.deepcell.org/)[30]. This dataset includes nuclear and cellular segmentations for each image; we used the cellular segmentations.

**LiveCell.** The LiveCell dataset consists of 3,188 training and 1,516 test images of eight different cell lines collected using phase-contrast microscopy (https://sartorius-research.github.io/LIVECell/)[29]. The images were segmented with overlaps allowed across masks. The Cellpose model cannot predict overlapping masks, so overlaps were removed, as described in the Cellpose 2.0 paper[35].

**Omnipose.** The Omnipose dataset includes two bacterial datasets, each with manual segmentations: fluorescent bacterial images (143 training and 75 test images) and phase-contrast microscopy bacterial images (249 training and 148 test images)[32].

**YeaZ.** The YeaZ dataset consists of two datasets of manually segmented yeast cells: phase-contrast images (16 2D training images and 6 2D test images) and bright-field images (229 training images and 77 test images)[34].

**DeepBacs.** The DeepBacs dataset consists of 155 training images and 35 test images of bacteria collected using bright-field microscopy and fluorescent imaging[33]. We excluded five test images when computing segmentation quality, because they were labeled in a different modality; this only had a small effect on the AP score.

**Reporting summary**
Further information on research design is available in the Nature Portfolio Reporting Summary linked to this article.

## Data availability
We generated a dataset of two-photon calcium imaging data for denoising available at https://doi.org/10.25378/janelia.27854442. We also used the *Drosophila* wing epithelia and *Tribolium* nuclei datasets from the CARE paper[16,64]. In terms of segmentation datasets, the 'cyto2' dataset is publicly available at https://www.cellpose.org/dataset and the other datasets were generated and shared by other laboratories[29–34,40–42].

## Code availability
Cellpose3 was used to perform all analyses in the paper. The code and GUI are available at https://www.github.com/mouseland/cellpose. Scripts for recreating the analyses in the figures are available at https://github.com/MouseLand/cellpose/tree/main/paper/3.0.

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

## Acknowledgements

C.S. and M.P. are funded by the Howard Hughes Medical Institute at the Janelia Research Campus. We thank M. Rariden for contributions to the Cellpose code base. We thank GENIE for creating the riboL1-jGCaMP8s mouse line.

## Author contributions

C.S. and M.P. designed the study, collected data, performed data analysis and wrote the paper.

## Competing interests

The authors declare no competing interests.

## Additional information

**Extended data** is available for this paper at https://doi.org/10.1038/s41592-025-02595-5.

**Correspondence and requests for materials** should be addressed to Carsen Stringer or Marius Pachitariu.

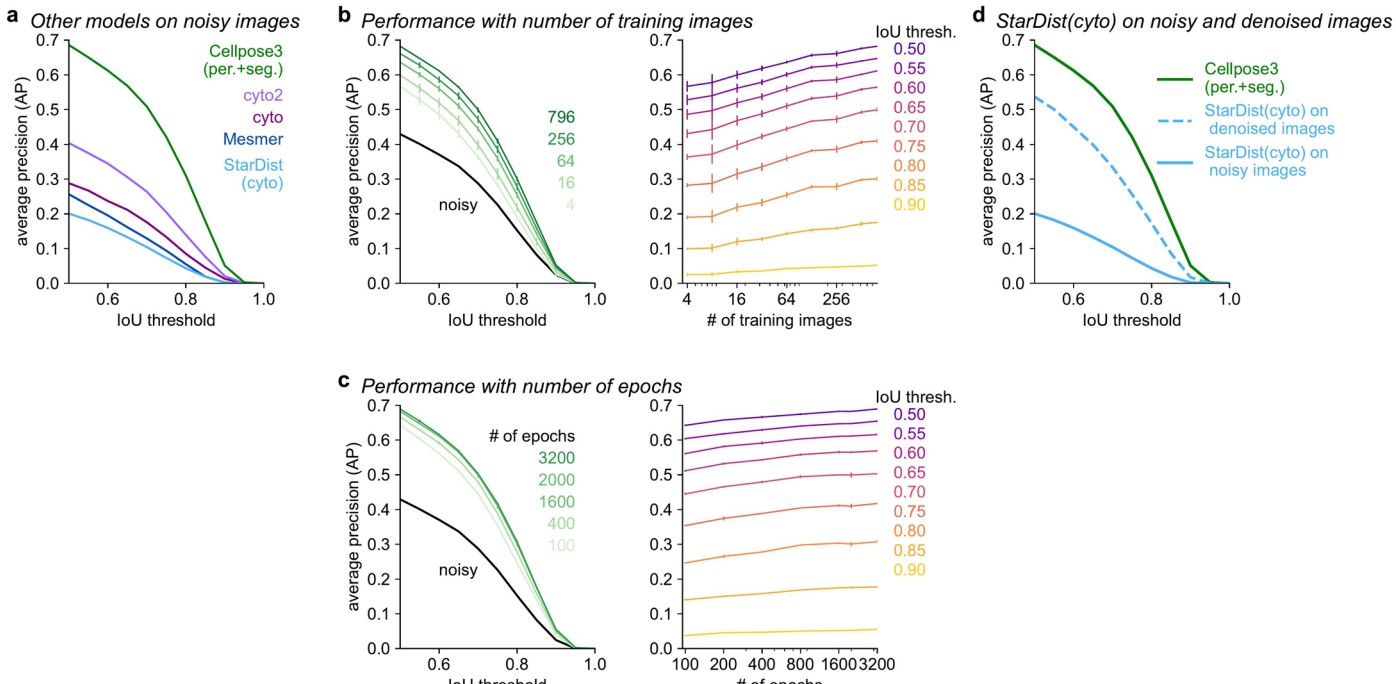

**Extended Data Fig. 1 | Performance for other models on noisy images and for Cellpose3 as a function of training parameters. a**, Segmentation performance of various models run on the noisy images in Fig. 1. 'cyto' and 'cyto2' size networks were used to estimate diameters from the noisy images for their segmentations. **b**, Cellpose3 networks trained with varying numbers of training images. Left: average precision curves for training on fixed image subsets. Right: same data

as Left, shown as a function of training images. n=5 random seeds, error bars represent s.e.m. **c**, Same as **b**, for networks trained with varying numbers of epochs. **d**, Segmentation performance of StarDist trained on the Cellpose cyto dataset, applied to noisy images (blue solid line) and applied to images denoised by the Cellpose3 (per.+seg. loss) network (blue dashed line). Segmentation with 'cyto2' (green) shown for comparison.

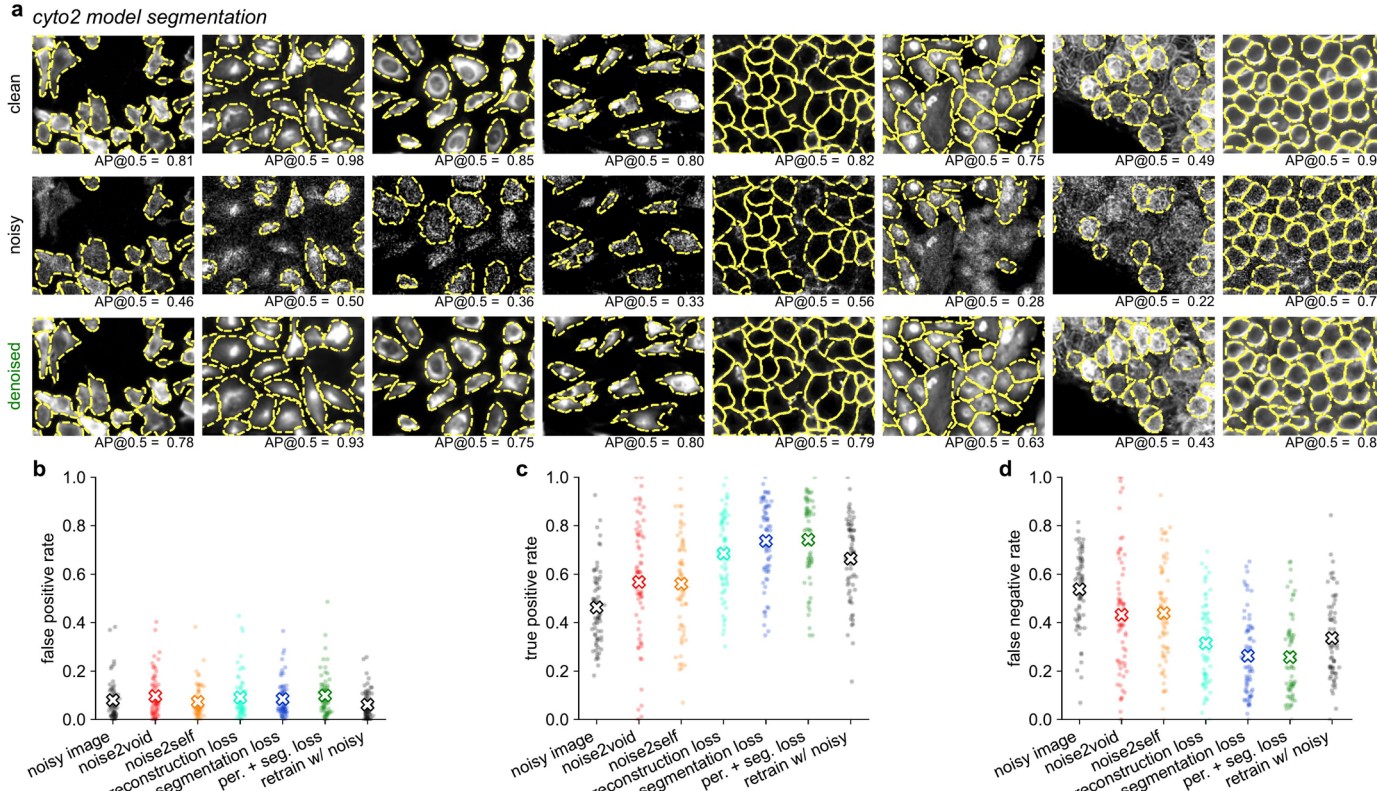

**Extended Data Fig. 2 | Segmentation of clean, noisy and denoised images of cells. a**, Same as Fig. 2a but overlaid with segmentations. **b**, The false positive rate of the segmentation at an IoU threshold of 0.5 for each of the 68 test images, either denoised and segmented with 'cyto2' or segmented directly with a model retrained on noisy images. Averages marked with "x". Wilcoxon two-sided signed-rank test between per. + seg. loss false positive rate and other results: p=0.089 (noisy), p=0.072 (Noise2Void), p=0.73 (Noise2Self), p=0.83 (rec. loss), p=0.013 (seg. loss), and p=1.7e-4 (retrain w/ noisy), n=68 test images. **c-d**, Same as **b** for the true positive rates and the false negative rates respectively.

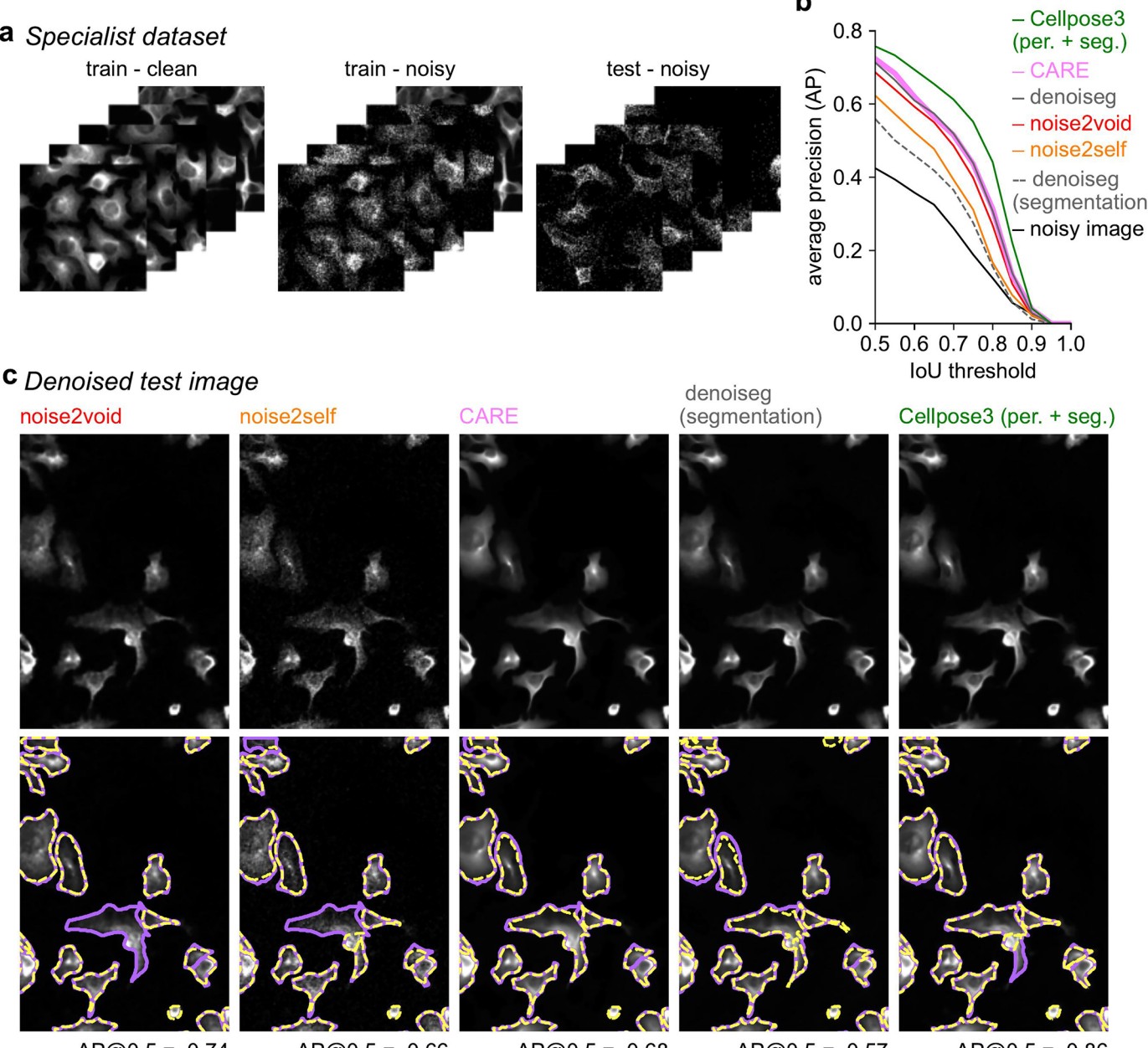

**Extended Data Fig. 3 | Denoising a specialist dataset. a**, 89 training images, clean and with Poisson noise added, and 11 test images with Poisson noise, from the CellImageLibrary CCDB:6843 dataset[39]. **b**, Mean AP score of segmentation applied to the noisy and denoised images, averaged across 11 test images. The CARE model was trained using the 89 noisy and ground-truth image pairs, and Noise2Self, Noise2Void, DenoiSeg were trained on the 100 noisy images from both the training and test set. DenoiSeg was also trained using the segmentations from the training images. The Cellpose3 model was the same as the model in Fig. 1, which was trained using all 'cyto2' images. All segmentations were performed using the 'cyto2' model, except for "DenoiSeg (segmentation)" which used the trained DenoiSeg network.

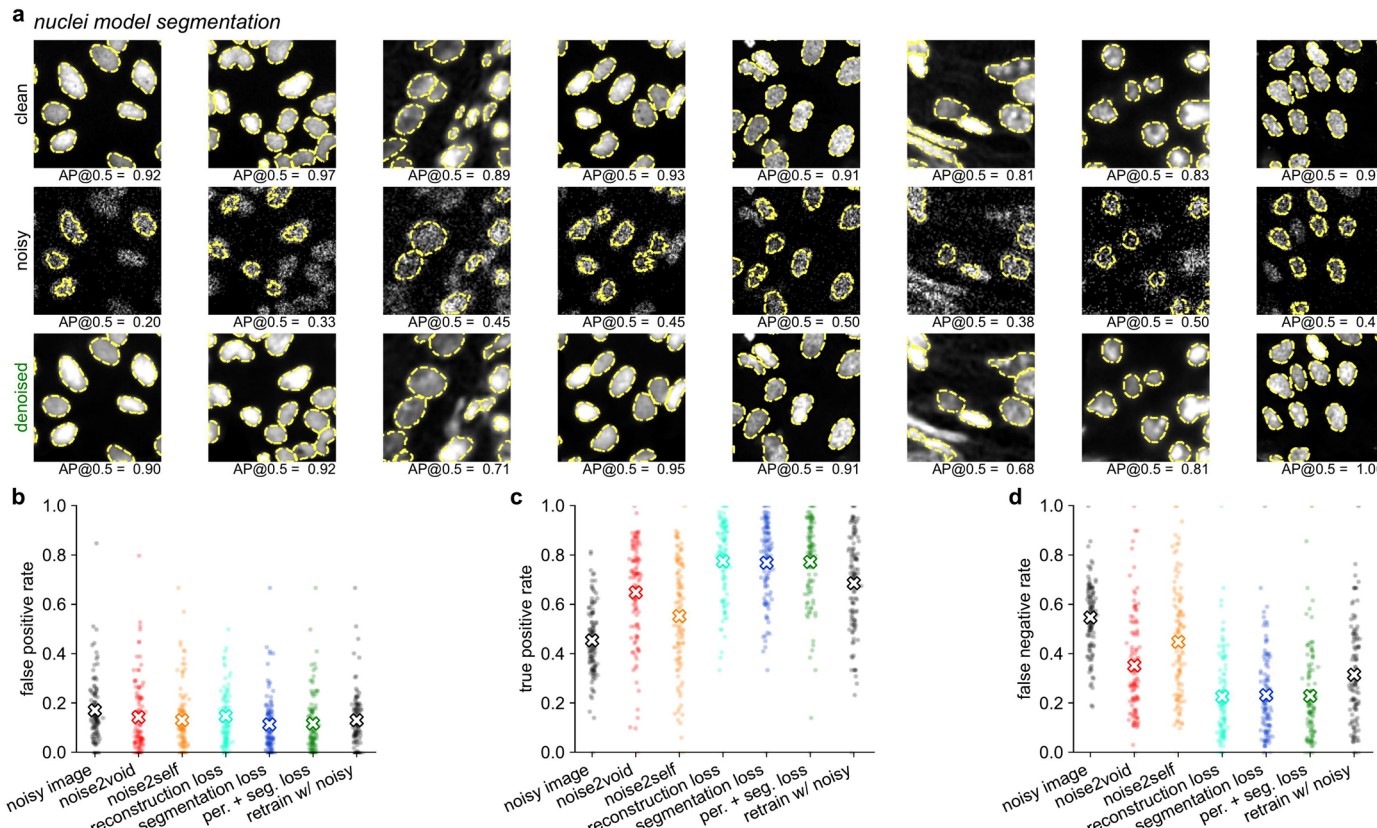

**Extended Data Fig. 4 | Segmentation of clean, noisy and denoised images of nuclei.** Same as Extended Data Fig. 2, for the Nuclei dataset. Wilcoxon two-sided signed-rank test between per. + seg. loss false positive rate and other results: p=3.6e-14 (noisy), p=1.1e-7 (Noise2Void), p=4.8e-8 (Noise2Self), p=1.3e-4 (rec. loss), p=0.45 (seg. loss), and p=8.6e-5 (retrain w/ noisy), n=111 test images.

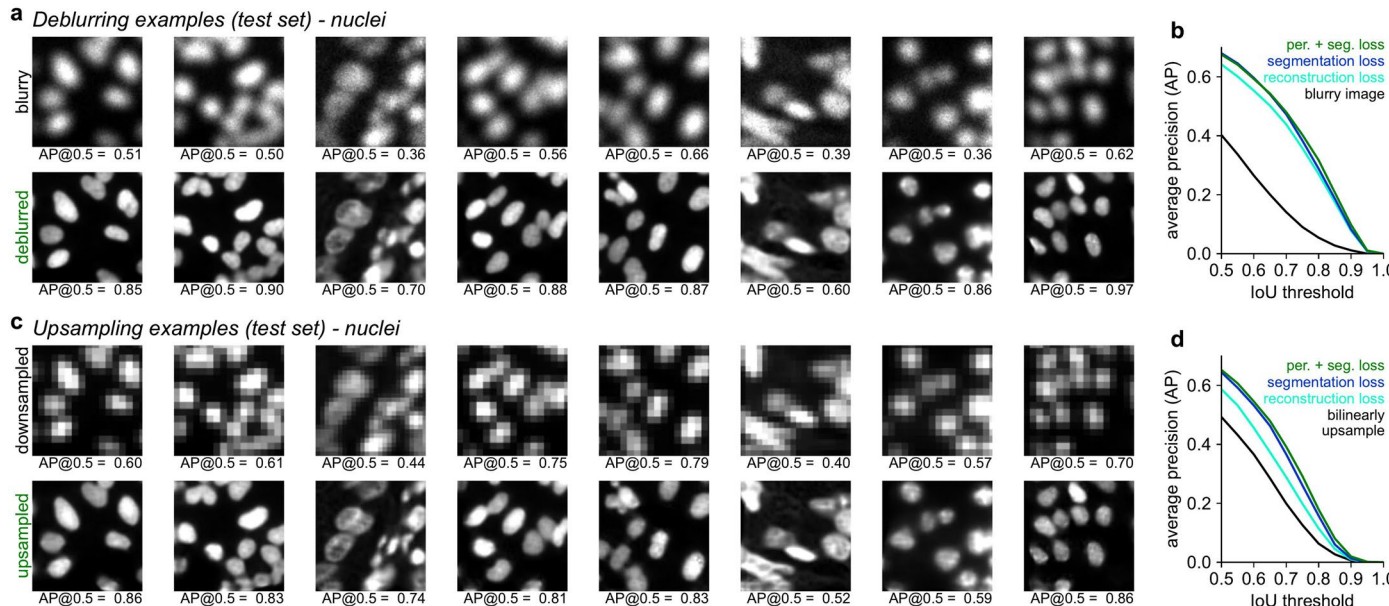

**Extended Data Fig. 5 | Deblurring and upsampling nuclear images.** Same as Fig. 4, for the Nuclei dataset.

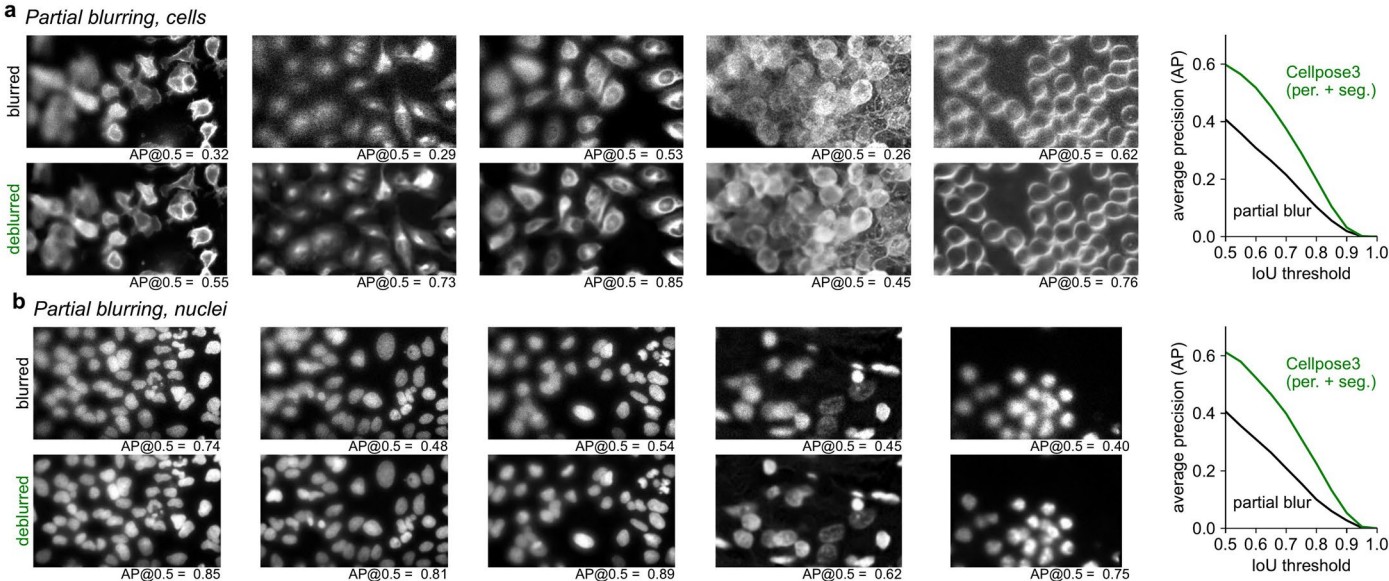

**Extended Data Fig. 6 | Deblurring images with partial blurring.** Cellpose3 deblurring for images with only half the image blurred. **a**, Same as Fig. 4a, with same deblurring and segmentation networks. **b**, Same as Extended Data Fig. 5a, with same deblurring and segmentation networks.

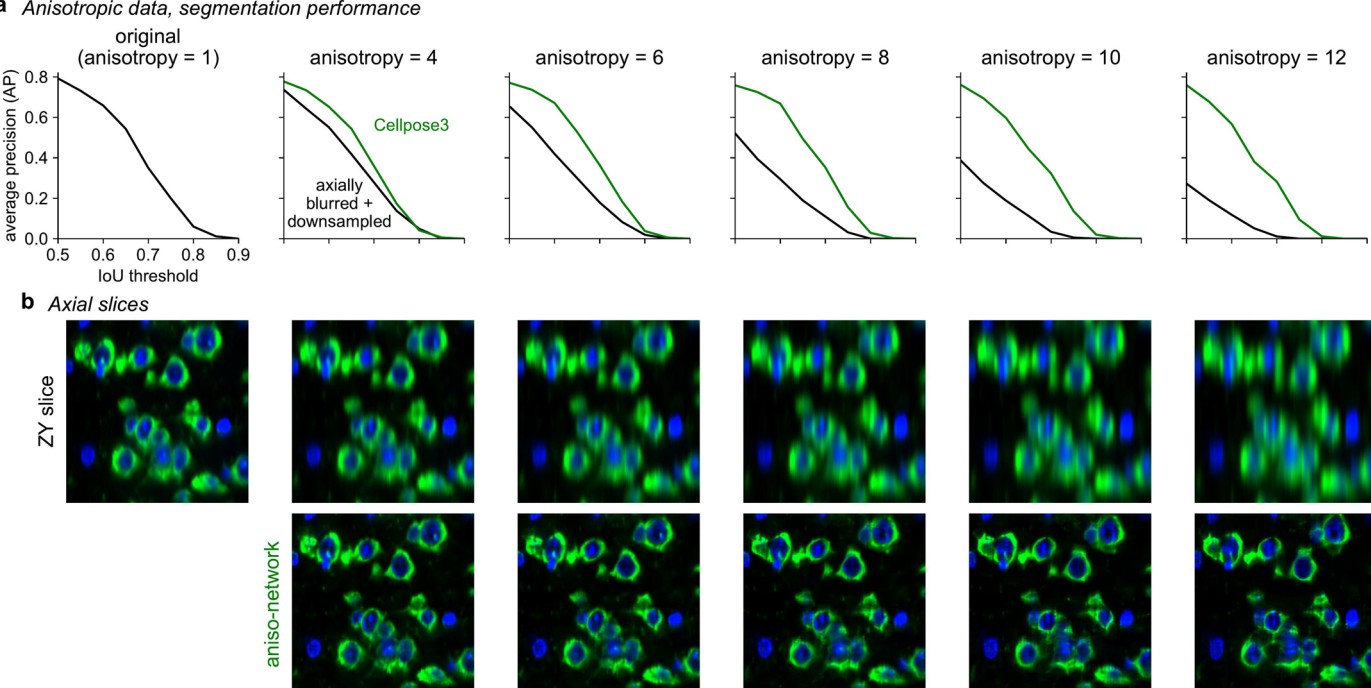

**a** *Anisotropic data, segmentation performance*

**b** *Axial slices*

**Extended Data Fig. 7 | Combined deblurring and upsampling for anisotropic datasets.** Cellpose3 networks were trained on the Cellpose and Nuclei datasets with axial blurring and downsampling. These networks were applied to a 3D dataset with ground-truth segmentation labels, with synthetic blurring and downsampling applied. **a**, Segmentation performance of 'cyto2' model before and after applying the Cellpose3 networks on the cellular and nuclear channels. **b**, Example axial slices from the volume, with blurring and downsampling (top), and deblurred/upsampled with the Cellpose3 network (bottom).

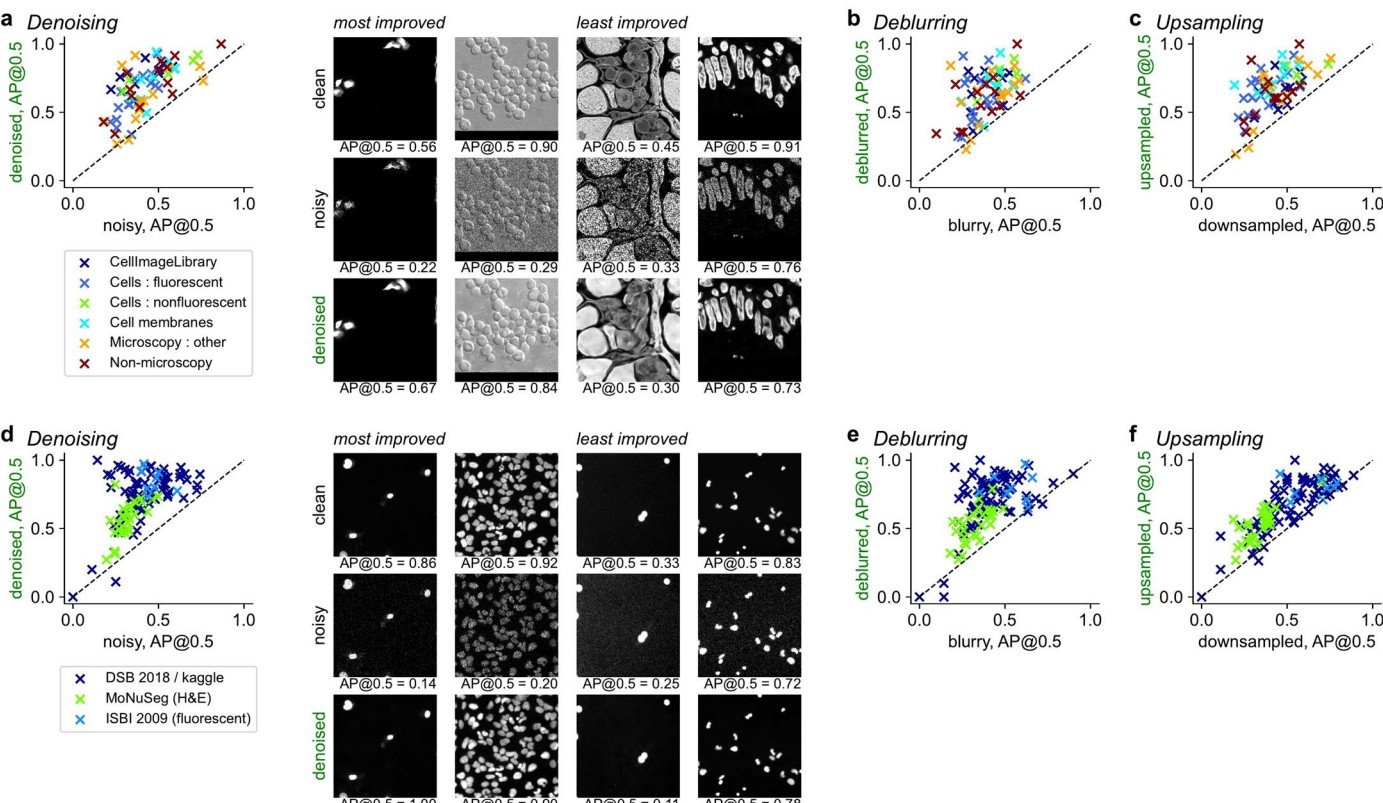

**Extended Data Fig. 8 | Most and least improved images after Cellpose3 restoration. a-c**, Denoising, deblurring and upsampling performance per image on the Cellpose test set, colored by image type. Most and least improved images from denoising shown in **a**. **d-f**, Same as **a-c** for the Nuclei test set.

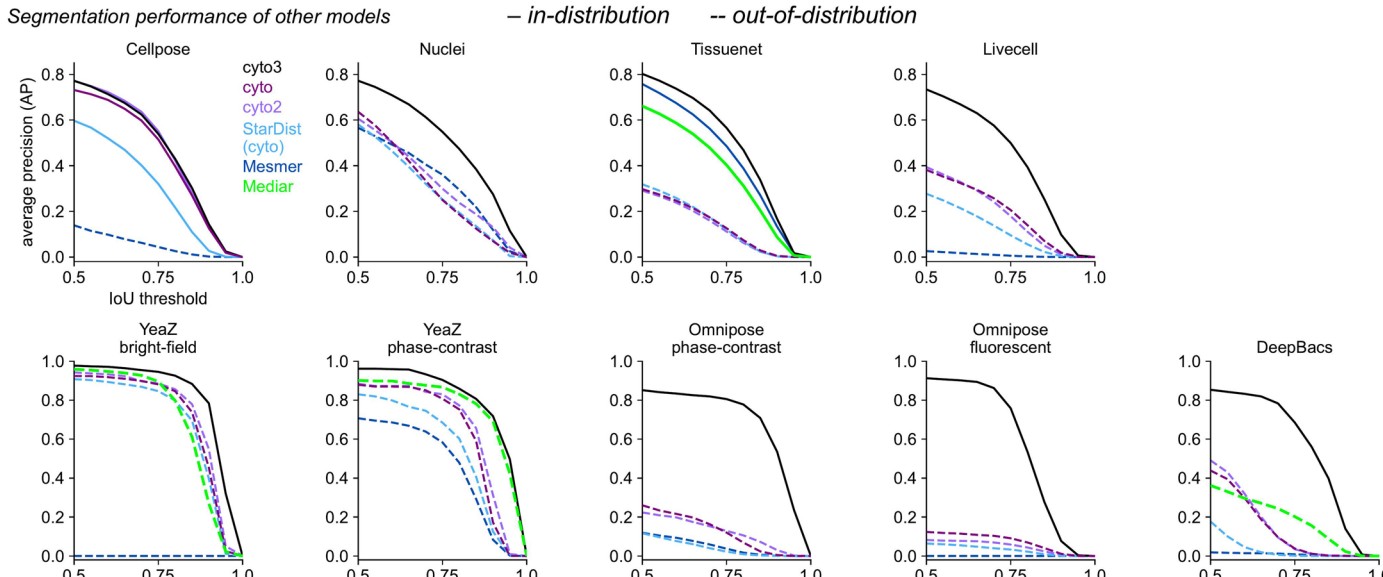

**Extended Data Fig. 9 | Out-of-distribution segmentation performance.**
We show "out-of-distribution" results for the three Cellpose models and three external segmentation models (StarDist, Mesmer and Mediar). "cyto" and "cyto2" were only trained on the Cellpose dataset, while cyto3 was trained on all datasets. StarDist was trained on the "cyto" Cellpose dataset. Mesmer was trained on the Tissuenet dataset to predict both cells and nuclei (nuclear prediction used for Nuclei test set). Mediar was trained on the Cellpose, Nuclei, Livecell and Omnipose datasets (both training and testing images) as well as the Neurips dataset which contained images acquired with the Tissuenet platform[68].

**a** *Deblurring examples from other datasets*

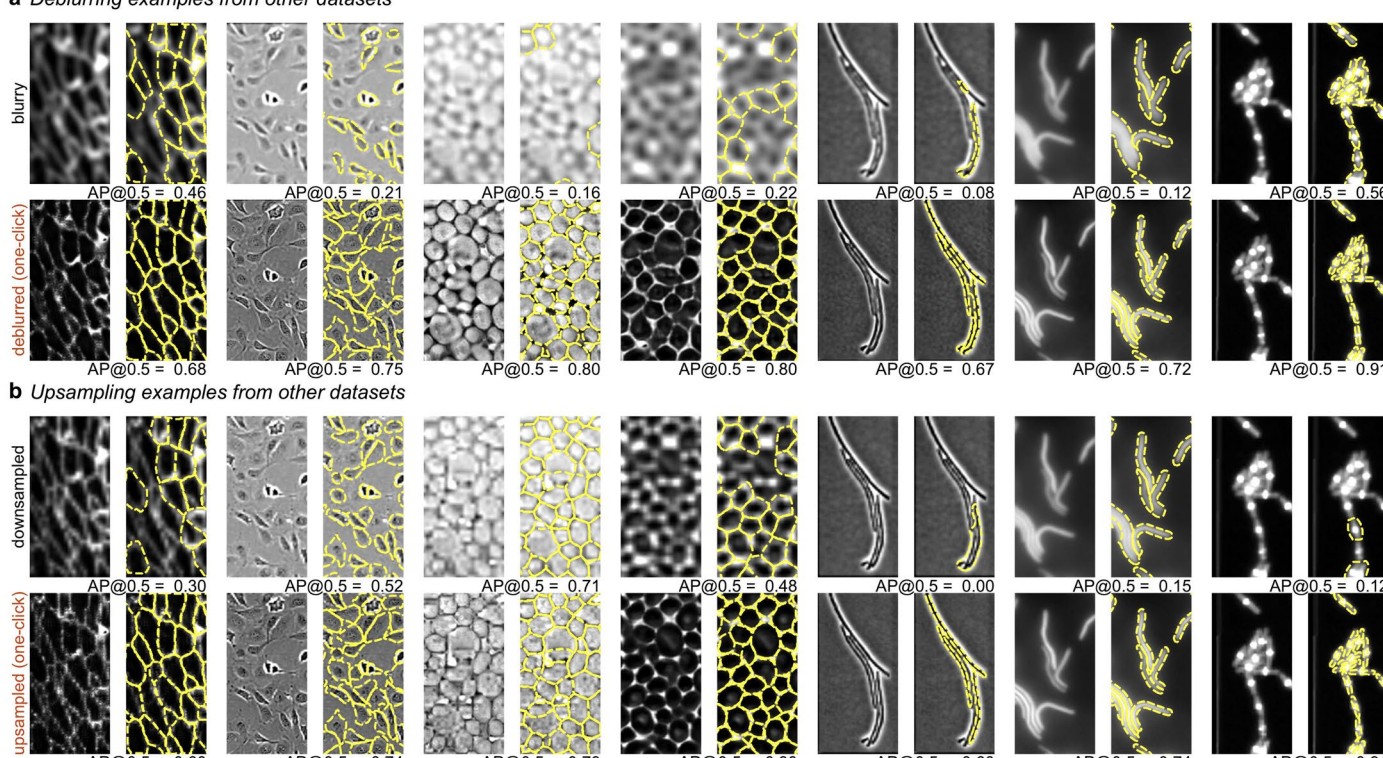

**b** *Upsampling examples from other datasets*

**Extended Data Fig. 10 | Examples of deblurring and upsampling on other datasets. a**, Example images and their segmentations from other datasets with Gaussian blurring (top) and deblurred with the one-click model (bottom), AP@0.5 reported (same one-click model as in Fig. 6d). **b**, Same as a for downsampled images, upsampled with the one-click model.

# Reporting Summary

## Statistics

For all statistical analyses, confirm that the following items are present in the figure legend, table legend, main text, or Methods section.

| n/a | Confirmed | |
|---|---|---|
| ☐ | ☒ | The exact sample size (*n*) for each experimental group/condition, given as a discrete number and unit of measurement |
| ☐ | ☒ | A statement on whether measurements were taken from distinct samples or whether the same sample was measured repeatedly |
| ☒ | ☐ | The statistical test(s) used AND whether they are one- or two-sided *Only common tests should be described solely by name; describe more complex techniques in the Methods section.* |
| ☒ | ☐ | A description of all covariates tested |
| ☒ | ☐ | A description of any assumptions or corrections, such as tests of normality and adjustment for multiple comparisons |
| ☒ | ☐ | A full description of the statistical parameters including central tendency (e.g. means) or other basic estimates (e.g. regression coefficient) AND variation (e.g. standard deviation) or associated estimates of uncertainty (e.g. confidence intervals) |
| ☒ | ☐ | For null hypothesis testing, the test statistic (e.g. *F*, *t*, *r*) with confidence intervals, effect sizes, degrees of freedom and *P* value noted *Give P values as exact values whenever suitable.* |
| ☒ | ☐ | For Bayesian analysis, information on the choice of priors and Markov chain Monte Carlo settings |
| ☒ | ☐ | For hierarchical and complex designs, identification of the appropriate level for tests and full reporting of outcomes |
| ☒ | ☐ | Estimates of effect sizes (e.g. Cohen's *d*, Pearson's *r*), indicating how they were calculated |

*Our web collection on statistics for biologists contains articles on many of the points above.*

## Software and code

Policy information about availability of computer code

| Data collection | Scanimage software v2022.1.0 (open source) was used to collect calcium imaging data from awake mice using a two-photon mesoscope (Thorlabs 2PRAM microscope). PsychToolbox was used to present visual stimuli during the experiments. |
|---|---|
| Data analysis | cellpose==3.0, csbdeep==0.7.4, efficientnet-pytorch==0.7.1,fastremap==1.14.0, imagecodecs==2023.9.18, imageio==2.31.6, jupyter==1.0.0, keras==2.14.0, matplotlib==3.8.0, n2v==0.3.2, numba==0.58.0, numpy==1.25.2, nvidia-cuda-runtime-cu11==11.8.89, nvidia-cuda-runtime-cu12==12.1.105, opencv-python-headless==4.8.1.78, python==3.9.18, scipy==1.11.3, segmentation-models-pytorch==0.3.3, tensorflow==2.14.0, tifffile==2023.9.26, timm==0.9.2, torch==2.1.0, torchvision==0.16.0, noise2self github (latest release, 2019) |

For manuscripts utilizing custom algorithms or software that are central to the research but not yet described in published literature, software must be made available to editors and reviewers. We strongly encourage code deposition in a community repository (e.g. GitHub). See the Nature Portfolio guidelines for submitting code & software for further information.

## Data

Policy information about availability of data

All manuscripts must include a data availability statement. This statement should provide the following information, where applicable:
- Accession codes, unique identifiers, or web links for publicly available datasets
- A description of any restrictions on data availability
- For clinical datasets or third party data, please ensure that the statement adheres to our policy

We generated a new clean/noisy dataset using two-photon calcium imaging, available at https://doi.org/10.25378/janelia.27854442. The 'cyto2' dataset is publicly available at https://www.cellpose.org/dataset, and the other datasets were generated and shared by other labs.

## Human research participants

Policy information about studies involving human research participants and Sex and Gender in Research.

| | |
|---|---|
| Reporting on sex and gender | N/A |
| Population characteristics | N/A |
| Recruitment | N/A |
| Ethics oversight | N/A |

Note that full information on the approval of the study protocol must also be provided in the manuscript.

# Field-specific reporting

Please select the one below that is the best fit for your research. If you are not sure, read the appropriate sections before making your selection.

☒ Life sciences          ☐ Behavioural & social sciences          ☐ Ecological, evolutionary & environmental sciences

For a reference copy of the document with all sections, see nature.com/documents/nr-reporting-summary-flat.pdf

# Life sciences study design

All studies must disclose on these points even when the disclosure is negative.

| | |
|---|---|
| Sample size | The sample size in this study was nine large datasets of cellular and nuclear images (2D), which each consisted of several classes of images from different cell types, tissues or imaging modalities. This dataset size was determined by the availability of large-scale fully annotated datasets: it is a substantial effort to create these datasets. These datasets were sufficient for determining the performance of models because they spanned imaging modalities and cell types with various morphologies. |
| Data exclusions | We excluded 5 test images from DeepBacs which were labeled in a different image modality (defined in Methods). |
| Replication | All the models were trained on large datasets (~800 cyto images, or ~1000 nuclei images, or ~8000 images across all nine datasets), and the training of each model takes up to 24 hours on an A100 GPU, so we did not perform multiple training replications for most analyses. We think this is reasonable because the datasets are so large. All attempts at replication were successful (e.g. we ran multiple seeds for Figure S1 which attained similar performance). |
| Randomization | There was no splitting of samples or organisms in this study to perform comparisons of experimental groups. |
| Blinding | There was no splitting of samples or organisms in this study to perform comparisons of experimental groups, so blinding is not applicable to this study. |

# Reporting for specific materials, systems and methods

We require information from authors about some types of materials, experimental systems and methods used in many studies. Here, indicate whether each material, system or method listed is relevant to your study. If you are not sure if a list item applies to your research, read the appropriate section before selecting a response.

## Materials & experimental systems

| n/a | Involved in the study |
|---|---|
| ☒ | ☐ Antibodies |
| ☒ | ☐ Eukaryotic cell lines |
| ☒ | ☐ Palaeontology and archaeology |
| ☐ | ☒ Animals and other organisms |
| ☒ | ☐ Clinical data |
| ☒ | ☐ Dual use research of concern |

## Methods

| n/a | Involved in the study |
|---|---|
| ☒ | ☐ ChIP-seq |
| ☒ | ☐ Flow cytometry |
| ☒ | ☐ MRI-based neuroimaging |

# Animals and other research organisms

Policy information about studies involving animals; ARRIVE guidelines recommended for reporting animal research, and Sex and Gender in Research

| | |
|---|---|
| Laboratory animals | riboL1-jGCaMP8s mouse, sex female, aged 4 months. |
| Wild animals | No wild animals used. |
| Reporting on sex | Sex-based analyses not required - neural activity not analyzed. |
| Field-collected samples | No field-collected sample. |
| Ethics oversight | IACUC board at HHMI Janelia Research Campus approved the research in the study. |

Note that full information on the approval of the study protocol must also be provided in the manuscript.

