## [Peer Review File · Nature Methods]

Cellpose3: one-click image restoration for improved cellular segmentation

Corresponding Author: Dr Carsen Stringer

Version 0:

Decision Letter:

26th Mar 2024

Dear Carsen,

Your Article, "Cellpose3: one-click image restoration for improved cellular segmentation", has now been seen by three reviewers. As you will see from their comments below, although the reviewers find your work of considerable potential interest, they have raised a number of concerns. We are interested in the possibility of publishing your paper in Nature Methods, but would like to consider your response to these concerns before we reach a final decision on publication.

We therefore invite you to revise your manuscript to address these concerns. The two concerns that stood out to us are (1) does this work on realistic data with different types of artifacts? and (2) is benchmarking against other tools. We ask that you focus your revision on addressing these two points to show general applicability and that Cellpose3 represents the state-of-the-art.

Regarding the second point, reviewer 1 requested a specific method be compared, but reviewer 3 was vague. I think for us, we're quite curious how CellPose3 will compare to Foundation models and other large generalist models. In fact, we just published today this cell segmentation competition paper, in which a transformer-based model was the overall winner (<https://www.nature.com/articles/s41592-024-02233-6>). Is there any way you could do some benchmarking against this winning model? It was not specifically requested by the reviewers, but it could be very useful for readers. This is not a mandate, and I'm happy to discuss this point. We do appreciate that Cellpose offers many advantages to users beyond pure performance.

Link Redacted

We hope to receive your revised paper within three months. If you cannot send it within this time, please let us know. In this event, we will still be happy to reconsider your paper at a later date so long as nothing similar has been accepted for publication at Nature Methods or published elsewhere.

OPEN SCIENCE REQUIREMENTS

REPORTING SUMMARY AND EDITORIAL POLICY CHECKLISTS

DATA AVAILABILITY

All novel DNA and RNA sequencing data, protein sequences, genetic polymorphisms, linked genotype and phenotype data, gene expression data, macromolecular structures, and proteomics data must be deposited in a publicly accessible database, and accession codes and associated hyperlinks must be provided in the "Data Availability" section.

CODE AVAILABILITY

Please include a "Code Availability" subsection in the Online Methods which details how your custom code is made available. Only in rare cases (where code is not central to the main conclusions of the paper) is the statement "available

upon request" allowed (and reasons should be specified).

MATERIALS AVAILABILITY

ORCID

Sincerely,
Rita

Rita Strack, Ph.D.
Senior Editor
Nature Methods

Reviewers' Comments:

Reviewer #1:

Remarks to the Author:

Summary of paper :

Cellpose3 addresses the problem of improving the segmentation of pre-trained models (in particular Cellpose models) given input images that are degraded by noise of a designated type.

Standard approaches either 1) denoise the input image via classical methods or self-supervision trained neural networks e.g. Noise2void, Noise2noise, Noise2self, or 2) supervision trained neural networks with clean target images e.g. CARE; or 3) by jointly training the segmentation network with a denoising/reconstruction loss and a classical segmentation loss (like Jaccard loss) e.g. denoiSeg.

Cellpose3 follows the modern data-driven neural network approach and proposes to formulate a 'sequential' problem whereby a pretrained Cellpose model functions as the 'segmentation loss' instead of a generic loss and to subsequently train a lightweight denoising network to transform the input image so as to improve the pretrained Cellpose segmentation losses.

Cellpose3 demonstrates its proposal on two types of artefacts commonly found in microscopy:

1. Microscope noise, modelled as Poisson or Gaussian noise arising often due to low laser power, depth scattering or weak staining
2. Blur noise modelled as Gaussian blur or due to upsampling/downsampling and shows improved performance on diverse 2D datasets where the artefacts are simulated except for one which was acquired at different laser powers and previously published, compared to either of:
 - i) Denoising first with self-supervised neural nets, then segment with Cellpose,
 - ii) Retraining the Cellpose segmentation model from scratch using the noisy images as input and segmentation labels associated with the clean images.

Overall Impression:

The new Cellpose3 approach (increasing applicability via learned image restoration preprocessing) extends Cellpose1 (establishment of a generalist algorithm) and Cellpose2 (increasing applicability via human-in-the-loop training). I find the approach overall interesting, and a simple method to fine-tune segmentation that is as far as I know, novel to biological cell segmentation. The paper is well-presented with a logical flow of experiments. The methodology and experimental design is overall sound, and the results appear promising. Cellpose cyto3 in particular appears to be a major step-up compared to previous versions and will be valuable to the community.

However, I have some concerns, comments and questions relating to the broad applicability of the denoising in Cellpose3 for users when using Cellpose3 on real datasets they would collect. In particular real datasets don't just display the artefact types demonstrated here: blur or shot/speckle noise. If they did, we wouldn't want to do quantitative analysis anyway and instead redo the experiment or optimize the acquisition and experimental conditions.

Comments / Queries :

Missing control experiments

1. There does not appear to be a direct comparison of the Cellpose3 finetuning approach in comparison to the setup of joint image segmentation + restoration loss by e.g. denoSeg (ref. 17) either from the pretrained weights or from scratch. This is an important missing control, particularly considering denoSeg does not require all images to have segmentation labels.

2. It is not clear the effectiveness of the Cellpose3 training setup if applied to train other pretrained segmentation models outside the Cellpose family of models, example.g. non-flow based segmentation models such as UNets, Mesmer, StarDist etc.

3. As the main metric is AP and evaluation is always on an in-distribution balanced val set, it is not clear the trained denoiser can now act on out-of-distribution images (e.g. in-house collected/ computer vision) as an image denoiser or is it really just functioning as an adaptive transformation layer and will not generalize to images of objects or modalities outside of the training? E.g. Can the trained denoiser network improve the segmentation of other pretrained segmentation models without further training or is it model-specific?

Model training and generalization

4. After training the denoiser, does segmentation performance further improve if the pretrained Cellpose segmentation model is unfrozen and allowed to fine-tune either by fixing the learnt denoising model or if jointly fine-tuning both? How much data is required to stably train the denoiser? What is the minimum number of training iterations to converge to a stable solution? Could a small number of 'diverse' examples be sufficient?

5. How effective and consistent is the Cellpose3 strategy when applied after training to segment 'sequential' image datasets e.g. a 2D timeseries or 3D acquired stack?

6. Training of the 'cyto3' super-generalist model. The importance of balancing the training dataset to account for dataset diversity was emphasized. However it is not clear what objective criteria if any that led to the relative % sampled from each dataset or if these were obtained by cross-validation and search. E.g. how was this optimal?

Applicability to artefacts in real-life datasets

7. Extension and applicability to other types of artefacts. e.g. motion blur, or anisotropy: 3D microscopy often has reduced axial resolution. Can the trained blurring denoising network improve segmentation on the xz, yz slices if interpolated to isotropic resolution? What about degradation due to non-uniform illumination commonly found in tissue? What type of artefacts might Cellpose3 not work on?

8. Real datasets comprise a composite of artefacts outside of the shot and blur noise here which are more detrimental such as uneven illumination and focus in tissue or debris/oversaturation/underexposure. The authors note, however, "that we still train individual models for each of the three restoration tasks; combining tasks into a single model was in fact detrimental to performance (results not shown)". – This seems a crucial discussion point and appears to be a strong limitation of Cellpose3. Could the authors present the data and discuss why this might be the case? e.g. is it multimodality, model capacity? How would the authors recommend using the different independently trained denoising models to address the presence of more than one artefact type in the image?

9. Since the denoiser appears to be specific for artefact type. Is there 'one-click' retraining for end-users allowing them to retrain for other artefacts?

Implications of results in specific figures

10. Fig. 1d and Fig. 1f. Not all images improve in segmentation performance with Cellpose3. The spread of the points are similar between noise2self/noise2void vs Cellpose3. What images have the largest improvement of the denoiser and is it possible to predict beforehand? For those that don't improve is this because they have a different noise distribution from that being modelled?

11. Fig.S1 and S3 suggests there is no detriment to mean performance on clean images and everything to gain if the artefact type (e.g. blur) is present when a denoiser is trained. Should users always adopt this approach when applying segmentation models for all real datasets?

Comments on Code :

I was able to verify the install of the GUI and code from the provided Github on both Windows (Python 3.9 and PyQt6) and Red Hat Enterprise Linux Server (Python 3.9, PyQt5 only) and use the GUI to load and segment 2D images.

Minor comments:

Lines 37-41 : "Segmentation of noisy images has been used as an auxiliary task for denoising, but the corresponding cost functions were fully separated and trained with different neural networks [17]".

The above statement is imprecise and incorrect with respect to the cited reference 17, DenoiSeg. In ref 17, a single UNet was trained to jointly denoise and segment using a loss function that is a weighted average of the reconstruction and standard segmentation loss. When segmentation labels were not available, the segmentation loss was switched off. Instead of describing as fully separated, it would be more standard to describe as a joint loss. What I see separates Cellpose3 is the training of a separate adaptive denoise network whose output must be sequentially further evaluated by the pretrained segmentation model.

Yours sincerely,
Dr. Felix Zhou

Reviewer #2:

Remarks to the Author:

General comments

- The authors address an important issue, namely cell segmentation in challenging samples which most image analysts are struggling with. Overall, the work is well-written and analyses well-prepared, the methods novel, citations appropriate, and colleagues have done an important service to the community by both improving the Cellpose algorithms even further and publishing models to the general public ensuring their wider use.

Major comments

1. My main critique is that after reading the manuscript and testing the models, I am not yet convinced that the algorithm works robustly on challenging real-world images. The reason might be that the synthetic artefacts created with Gaussian or Poisson noise might not reflect patterns observed naturally or that the training configurations do not yield sufficient benefit. I also tested all restoration types on fluorescence images of human cells (immunohistochemistry staining) with noise or blurring but I could personally distinguish individual cells.

2. Although the authors presented the results well, I find the inability to integrate all three restorations in one model disappointing. Image datasets might include multiple types of degradations. How is Cellpose3 supposed to be used? Should multiple restorations be used or just one? Although possibly challenging to implement, could there be a preliminary algorithm measuring whether there is and the type of degradation, and would then select the correct restoration? For instance, a CNN classifying images to normal vs. noisy vs. blurred vs. downsampled, and then proceed with the optimal restoration if needed. Keep in mind that the noise might cover just a small part of the image and not the whole image as in the examples used in this manuscript.

Minor comments

Title, abstract and introduction

3. The authors have done an excellent job, no critical comments

Results

4. L64-67 "This may be due to sensor noise, which is often described as Gaussian, or shot noise which is well captured by a Poisson distribution." and L74-77 "Since Poisson noise becomes Gaussian-like in the limit of high simulated brightness, this approach should model both the sensor noise and the shot noise."

- Please provide evidence of the first sentence and evidence that Poisson noise capture the naturally occurring noise in the samples. Visually, the noise does not reflect the type of noise naturally seen in the images.

5. L159-163 "We found that the false positive rates were similar across all denoising approaches and similar to the false positive rate of the original noisy images (Figure S1b-d).

- I agree with the statement when visually examining the plots. Consider providing statistical evidence, for example with p-values of a T-test (or Wilcoxon rank-signed test). Tests are required only for the cellpose3 results vs. the others.

6. L187-189 "In addition, we trained CARE using pairs of noisy and clean training images from the training data of the specialized dataset"

- Consider adding a reference to the methods, e.g. "See methods" or similar as it is not evident for everyone what CARE is.

7. L189-190 "Cellpose3 also outperformed this approach."

- Add mention to the Figure S2b,c.

8. L199-201 "We suspect this is because nuclear segmentation relies on simpler object shapes which may be easier to denoise."

- Do the authors see any possible pitfalls in testing the Cellpose3 in a dataset the Cellpose2 was trained on? Could this also influence that there are less differences between the different losses?

9. L223-224 "For deblurring, we trained our networks on images with Gaussian blur of random spatial sizes."

- Why Gaussian blur as you used Poisson blur before? To my understanding, Gaussian blur does not perfectly reflect

naturally occurring blur.

10. L385 "Each network was trained for 2000 epochs"

- What was the rationale for 2000? Would the results be better by controlling training with checkpoints based on minimizing the diverse loss functions presented in this manuscript?

11. How well do the restorations work for brightfield images? Could the authors include brightfield (e.g. H&E-stained or giemsa-stained) cell images to demonstrate if the algorithms work with these?

Discussion

12. In case some of the issues raised above could not be addressed, these should be described in the Discussion part.

Reviewer #3:

Remarks to the Author:

In this manuscript, Carsen et al., proposed an improved strategy for cellular segmentation. In this strategy, the developed method addressed the inherent issues associated with degraded microscopy images, such as noise, blurring, and undersampling. Instead of directly estimating from the noisy images, the authors employed a denoising network to restore the image information, and subsequently utilized a segmentation network to extract cellular information. The approach demonstrated improved performance through a series of simulated experiments and selected real experiments. However, it is worth noting that the denoising network component of the developed method is derived from existing work, and the segmentation network is based on their previously published work, which suggests a relatively moderate level of novelty in the approach. Furthermore, the majority of the data utilized in this study involved augmenting existing datasets with noise, blur, and undersampling. Only Fig. 2b and 3c were obtained from actual acquired data. Additionally, the manuscript lacks a comparison with other state-of-art segmentation methods. The author only compared the impact of their own segmentation method with and without denoising, without providing a comprehensive evaluation against other existing approaches.

According to the above consideration, I cannot recommend publication the manuscript.

Besides, there are other minor points to be addressed:

1. Inaccurate label. For example, in Fig. 1c, the authors used 'noise2void' to describe left image. However, the figure caption was described as 'denoised with noise2self (left), and segmented (right)'.

2. Inconsistent performance. In Fig. 1j, the average precision results showed that there was not a significant difference between using segmentation loss and perceptual + segmentation loss, retrain with noise performed better than using only reconstruction loss. However, in Fig. 3b, the average precision results indicated that the three losses, including reconstruction loss, segmentation loss and perceptual + segmentation loss, showed similar performance. Notably, the reconstruction loss significantly outperformed the retrain with noise approach. This suggests that the performance of using reconstruction loss fluctuates significantly, and there is not a substantial difference between using segmentation loss and perceptual + segmentation loss.

3. In figure 4 (d) on page 5, the average precision curve with perceptual and segmentation loss appears to be the same as the average precision curve with segmentation loss. Why does perceptual loss hardly work for undersampled cellular images? In addition to average precision curves, all experimental results should also be listed in tables.

4. As noted in line 256, individual models are trained for each of the three restoration tasks, and a single model was detrimental to performance. However, real biological images may be simultaneously degraded by noise, blurred, or undersampled.

Version 1:

Decision Letter:

Our ref: NMETH-A55519A

10th Oct 2024

Dear Carsen,

Thank you for submitting your revised manuscript "Cellpose3: one-click image restoration for improved cellular segmentation" (NMETH-A55519A). It has now been seen by the original referees and their comments are below. The reviewers find that the paper has improved in revision, and therefore we'll be happy in principle to publish it in Nature Methods, pending minor revisions to satisfy the referees' final requests and to comply with our editorial and formatting guidelines.

We do not ask for any further experiments in response to the referee concerns, but we do ask that you provide a point-by-point rebuttal reflecting your responses and any associated changes to the main text.

TRANSPARENT PEER REVIEW

ORCID

Sincerely,
Rita

Rita Strack, Ph.D.
Senior Editor
Nature Methods

Reviewer #1 (Remarks to the Author):

Thanks to the authors for taking the time to revise the manuscript. I appreciate the addition of the various new validation experiments; the scaling of the training, application to 3D, inhomogeneous blurring, comparison to other segmentation methods with/without cellpose3 denoising, and the revised and improved 'oneclick' denoising model, (Figure 6). I think the manuscript is greatly improved, addresses the review comments and the method overall now much more convincing and enticing for end-users, particularly with the 'oneclick' restoration model.

I look forward to using Cellpose3 denoising.

Some minor comments of the revised paper:

- 1) Fig.6b - black line needs labeling in legend + figure for readers. I assume this is the respective base model (cyto2 or nuclei) without denoising.
- 2) Fig. 6d - I found it weird to show visual results for added poisson noise but not for other noise types in the same figure. Is this a typo? as the row label is 'noisy'. If not, it would make sense to also show the other noise types i.e. for deblur and upsampling.
- 3) Fig S3C - for completeness and transparency, I recommend including denoiSeg segmentation visualization results so all models plotted in the AP curves is represented.
- 4) Fig S9 - the double dashed lines for out-of-distribution is too small in the legend. It looked on first glance identical to the single line. I recommend increasing the size, and including an additional note in the figure legend to clarify, particularly since the 'in-distribution' dataset is different for individual models with the sole exception of cyto3.

Felix Zhou

Reviewer #1 (Remarks on code availability):

I have installed and verified the python library, and verify that the code API works for running Cellpose3 denoising model and the new 'oneclick' model on the provided image data. The new cellpose3 models and functionality has been documented and authors have included jupyter notebooks for running which is very helpful and I have verified they do work.

I have previously verified installation of the GUI.

Reviewer #2 (Remarks to the Author):

The authors addressed all my prior comments and concerns with additional data and analyses.

Reviewer #2 (Remarks on code availability):

I was able to successfully use the code. The instructions provided in the code repository were clear and the results reflect sufficiently the description provided in the manuscript.

Reviewer #3 (Remarks to the Author):

Although I am certain that the as-described method will be useful in the cellular segmentation community, I do not think that this work will be as crucial as their two previously published works, either I do not think its importance matches those expected for a journal like Nature Methods.

Some of contributions in the article have already been proposed by existing work, like the combination of restoration and segmentation [1,2], perceptual loss in image restoration [3]. I believe that the authors should and have the ability to pay more attention on the breadth and diversity of training dataset as stated in line 337-339, the core of generalist models.

[1]Paul G, Cardinale J, Sbalzarini I F. Coupling image restoration and segmentation: A generalized linear model/Bregman perspective[J]. International journal of computer vision, 2013, 104: 69-93.

[2]Niu X, Yan B, Tan W, et al. Effective image restoration for semantic segmentation[J]. Neurocomputing, 2020, 374: 100-108.

[3]Zhao H, Gallo O, Frosio I, et al. Loss functions for image restoration with neural networks[J]. IEEE Transactions on computational imaging, 2016, 3(1): 47-57.

Authors emphasizes that the proposed method is essentially an image restoration approach. However, the comparison with other methods is unfair. For example, the segmentation information used by Cellpose 3 originates from GT images, which definitely including GT content. On the contrary, unsupervised image restoration methods such as Noise2Noise and Noise2self only use the training data itself.

In practical scenarios, microscopic images often require manual screening or localization of degraded areas, where many degradations are scattered and localized. As a "one click" model, the authors did not present the results of a real and complete image, nor did they explain the impact of clear images passing through the network.

Reviewer #1:

Remarks to the Author:

Summary of paper :

Cellpose3 addresses the problem of improving the segmentation of pre-trained models (in particular Cellpose models) given input images that are degraded by noise of a designated type.

Standard approaches either 1) denoise the input image via classical methods or self-supervision trained neural networks e.g. Noise2void, Noise2noise, Noise2self, or 2) supervision trained neural networks with clean target images e.g. CARE; or 3) by jointly training the segmentation network with a denoising/reconstruction loss and a classical segmentation loss (like Jaccard loss) e.g. denoiSeg.

Cellpose3 follows the modern data-driven neural network approach and proposes to formulate a 'sequential' problem whereby a pretrained Cellpose model functions as the 'segmentation loss' instead of a generic loss and to subsequently train a lightweight denoising network to transform the input image so as to improve the pretrained Cellpose segmentation losses.

Cellpose3 demonstrates its proposal on two types of artefacts commonly found in microscopy:

1. Microscope noise, modelled as Poisson or Gaussian noise arising often due to low laser power, depth scattering or weak staining

2. Blur noise modelled as Gaussian blur or due to upsampling/downsampling

and shows improved performance on diverse 2D datasets where the artefacts are simulated except for one which was acquired at different laser powers and previously published, compared to either of:

- i) Denoising first with self-supervised neural nets, then segment with Cellpose,

- ii) Retraining the Cellpose segmentation model from scratch using the noisy images as input and segmentation labels associated with the clean images.

Overall Impression:

The new Cellpose3 approach (increasing applicability via learned image restoration preprocessing) extends Cellpose1 (establishment of a generalist algorithm) and Cellpose2 (increasing applicability via human-in-the-loop training). I find the approach overall interesting, and a simple method to fine-tune segmentation that is as far as I know, novel to biological cell segmentation. The paper is well-presented with a logical flow of experiments. The methodology and experimental design is overall sound, and the results appear promising. Cellpose cyto3 in particular appears to be a major step-up compared to previous versions and will be valuable to the community.

Thank you.

However, I have some concerns, comments and questions relating to the broad applicability of the denoising in Cellpose3 for users when using Cellpose3 on real datasets they would collect. In particular real datasets don't just display the artefact types demonstrated here: blur or

shot/speckle noise. If they did, we wouldn't want to do quantitative analysis anyway and instead redo the experiment or optimize the acquisition and experimental conditions.

There are many situations where such an experiment cannot be redone. We did not motivate this properly in the introduction, which was a lapse on our part and we rectified this in the revision with a new first paragraph (Lines 2-21).

Comments / Queries :

Missing control experiments

1. There does not appear to be a direct comparison of the Cellpose3 finetuning approach in comparison to the setup of joint image segmentation + restoration loss by e.g. denoiSeg (ref. 17) either from the pretrained weights or from scratch. This is an important missing control, particularly considering denoiSeg does not require all images to have segmentation labels.

Thanks for suggesting this analysis. We performed a direct comparison with DenoiSeg, including the train images with masks and the test images without masks, and we added the results to Figure S3b. This approach performed similarly to Noise2Void, which it is based on, and was outperformed by the generalist Cellpose3 network.

2. It is not clear the effectiveness of the Cellpose3 training setup if applied to train other pretrained segmentation models outside the Cellpose family of models, example.g. non-flow based segmentation models such as UNets, Mesmer, StarDist etc.

The Cellpose3 training setup could in principle be applied to other segmentation models. While potentially interesting, this is outside the scope of the present work because determining the best way to train these other methods with the Cellpose3 training setup is a complex task in itself, and would require bespoke code in other deep learning frameworks such as Tensorflow. The other methods are also not as good as Cellpose at segmentation, so while we could indeed improve their performance on noisy images, it would still not come close to what Cellpose can achieve.

We are continually benchmarking Cellpose against new models, for example see our response paper to Mediar, the winner of the Neurips challenge described in Ma et al, Nat Methods 2024 (Stringer and Pachitariu, "Transformers do not outperform Cellpose"). Cellpose outperforms Mediar, which is basically a transformer-based version of Cellpose, and even our own transformer-based implementation performs just as well as the original Cellpose model (Figure 5 of this manuscript). As for 2-class and 3-class U-nets, they perform quite poorly at cellular segmentation, as shown in the original Cellpose paper. Methods like Stardist and Mesmer perform better, but they still are significantly outperformed by Cellpose on cellular datasets (see Cellpose and Cellpose2 papers respectively, as well as the new Figure S9 in this manuscript).

3. As the main metric is AP and evaluation is always on an in-distribution balanced val set, it is not clear the trained denoiser can now act on out-of-distribution images (e.g. in-house collected/computer vision) as an image denoiser or is it really just functioning as an adaptive transformation layer and will not generalize to images of objects or modalities outside of the training?

Our evaluation was not always on an in-distribution balanced dataset. We used two external datasets for testing the method (Figure 2b,c and Figure 3c,d). These have the advantage that clean versions of the images were available, so we could define a ground truth segmentation. Unfortunately, there are very few datasets like these ones with both clean and noisy images. To

add to this, we collected an additional new dataset of two-photon imaging in vivo for testing purposes. Cellpose3 also generalized well to this use case (Figure 2d,e).

We should point out that true “out-of-distribution” generalization is rarely possible (see new Figure S9). This is similar to other machine learning domains, and is the reason why machine learning methods are trained on very large datasets that cover as much data diversity as possible. We have created such a dataset for the original Cellpose model, which has been sufficient for Cellpose to be useful broadly in the community. We have improved on that here by pooling together more datasets collected by other groups.

E.g. Can the trained denoiser network improve the segmentation of other pretrained segmentation models without further training or is it model-specific?

For this question, we tested the StarDist segmentation algorithm, and found that indeed segmentation quality improved after denoising with Cellpose3 (new Figure S1d). However, as mentioned in our comment above, StarDist performs worse than Cellpose at segmentation overall (see Cellpose1 paper), so the absolute level of performance after denoising is not the same as with Cellpose.

Model training and generalization

4. After training the denoiser, does segmentation performance further improve if the pretrained Cellpose segmentation model is unfrozen and allowed to fine-tune either by fixing the learnt denoising model or if jointly fine-tuning both?

Thanks for the suggestion. We tried this without success. Whenever both models are jointly trained or fine-tuned, we do not get the best restoration since the second network can compensate for imperfect restorations of the first one. We also do not observe better cell segmentation.

How much data is required to stably train the denoiser? What is the minimum number of training iterations to converge to a stable solution? Could a small number of ‘diverse’ examples be sufficient?

We have added an analysis to answer this question, along with the test accuracy as a function of the number of epochs (new Figure S1b,c).

5. How effective and consistent is the Cellpose3 strategy when applied after training to segment ‘sequential’ image datasets e.g. a 2D timeseries or 3D acquired stack?

Yes, there can be inconsistencies in the slice-by-slice denoising, similar to how there can be inconsistencies in the slice-by-slice segmentations. Despite those, the 2D-to-3D method in Cellpose generally works, but there may be cases that require some smoothing or end-to-end trained 3D denoising. We added a note about this in the text (Lines 294-298).

6. Training of the ‘cyto3’ super-generalist model. The importance of balancing the training dataset to account for dataset diversity was emphasized. However it is not clear what objective criteria if any that led to the relative % sampled from each dataset or if these were obtained by cross-validation and search. E.g. how was this optimal?

We did not attempt to optimize these probabilities. We simply used a larger proportion of the “cyto2” and “nuclei” datasets because they had the most diversity of all datasets, and we gave low probabilities to the Livecell and Tissuenet datasets, because those contain a large number of similar-looking images. Attempting to optimize these probabilities may not be ideal either: there is an attempt in the Mediar paper, but the resulting model still did not perform as well as Cellpose (as we showed in our recent response “Transformers do not outperform Cellpose”).

Applicability to artefacts in real-life datasets

7. Extension and applicability to other types of artefacts. e.g. motion blur, or anisotropy: 3D microscopy often has reduced axial resolution. Can the trained blurring denoising network improve segmentation on the xz, yz slices if interpolated to isotropic resolution?

Thanks for this suggestion, we trained new models for cells and nuclei to deblur and upsample axially (Lines 282-298). We applied these networks successfully in the new Figure S7 to data with ground-truth 3D segmentations.

What about degradation due to non-uniform illumination commonly found in tissue? What type of artefacts might Cellpose3 not work on?

For non-uniform illumination, we have an option to perform normalization of the image in a tiled fashion (“tile_norm_blocksize” parameter). Some artifacts may already be addressed in the original Cellpose model, since it was trained on images from realistic experiments. If this is found to be a more general problem, we could probably include non-uniform illuminations as an augmentation during training. We added an analysis to show that non-uniform blurring already works well without imposing special considerations (Figure S6).

8. Real datasets comprise a composite of artefacts outside of the shot and blur noise here which are more detrimental such as uneven illumination and focus in tissue or debris/oversaturation/underexposure. The authors note, however, “that we still train individual models for each of the three restoration tasks; combining tasks into a single model was in fact detrimental to performance (results not shown)”. – This seems a crucial discussion point and appears to a strong limitation of Cellpose3. Could the authors present the data and discuss why this might be the case? e.g. is it multimodality, model capacity? How would the authors recommend using the different independently trained denoising models to address the presence of more than one artefact type in the image?

Thanks for discussing this. We worked more on this problem, and eventually achieved success. One key aspect was to use a uniform rather than long-tailed distribution of blurring sizes. We present the results in Figure 6. This model is also now included in the GUI.

9. Since the denoiser appears to be specific for artefact type. Is there ‘one-click’ retraining for end-users allowing them to retrain for other artefacts?

These models can be trained using the command line interface and we have now documented the training commands. We do not plan to implement training of new denoising models inside the GUI, which would be a little too complex for normal users and would almost certainly not be just “one click”.

Implications of results in specific figures

10. Fig. 1d and Fig. 1f. Not all images improve in segmentation performance with Cellpose3. The spread of the points are similar between noise2self/noise2void vs Cellpose3. What images

have the largest improvement of the denoiser and is it possible to predict beforehand? For those that don't improve is this because they have a different noise distribution from that being modelled?

We did not see any obvious properties of the images which were well denoised or poorly denoised (or deblurred or upsampled). This is now presented in the new Figure S8. The variability is probably just randomness.

11. Fig.S1 and S3 suggests there is no detriment to mean performance on clean images and everything to gain if the artefact type (e.g. blur) is present when a denoiser is trained. Should users always adopt this approach when applying segmentation models for all real datasets?

Users always have the option to do this in the GUI and in the API. In practice, different users probably need different things, and we try to provide options for many different approaches rather than prescribing a single approach.

Comments on Code :

I was able to verify the install of the GUI and code from the provided Github on both Windows (Python 3.9 and PyQT6) and Red Hat Enterprise Linux Server (Python 3.9, PyQT5 only) and use the GUI to load and segment 2D images.

Great, thanks for checking the code.

Minor comments:

Lines 37-41 : "Segmentation of noisy images has been used as an auxiliary task for denoising, but the corresponding cost functions were fully separated and trained with different neural networks [17]"

The above statement is imprecise and incorrect with respect to the cited reference 17, DenoiSeg. In ref 17, a single UNet was trained to jointly denoise and segment using a loss function that is a weighted average of the reconstruction and standard segmentation loss. When segmentation labels were not available, the segmentation loss was switched off. Instead of describing as fully separated, it would be more standard to describe as a joint loss. What I see separates Cellpose3 is the training of a separate adaptive denoise network whose output must be sequentially further evaluated by the pretrained segmentation model.

Thank you, this was indeed incorrectly described. We have fixed it and we included direct comparisons with DenoiSeg as well (Figure S3b).

Yours sincerely,

Dr. Felix Zhou

Reviewer #2:

Remarks to the Author:

General comments

- The authors address an important issue, namely cell segmentation in challenging samples which most image analysts are struggling with. Overall, the work is well-written and analyses well-prepared, the methods novel, citations appropriate, and colleagues have done an important service to the community by both improving the Cellpose algorithms even further and publishing models to the general public ensuring their wider use.

Thank you for the positive feedback.

Major comments

1. My main critique is that after reading the manuscript and testing the models, I am not yet convinced that the algorithm works robustly on challenging real-world images. The reason might be that the synthetic artefacts created with Gaussian or Poisson noise might not reflect patterns observed naturally or that the training configurations do not yield sufficient benefit. I also tested all restoration types on fluorescence images of human cells (immunohistochemistry staining) with noise or blurring but I could personally distinguish individual cells.

We had shown good results on two datasets with experimental degradations, where ground truth images with no degradation were available (Figure 2b,c and Figure 3c,d). These were the only such datasets we could find that we could have used for validation. We have also now collected an additional fluorescence imaging dataset in our lab (Figure 2de) on which Cellpose3 also works well. We do not necessarily find this surprising, because the Poisson degradations should be very close in a mathematical sense to the effect of shot noise, which is due to the Poisson distributions of emitted photons from a sample. Similar arguments apply for blurring and downsampling.

We are unsure from this comment what results the reviewer obtained on the images they refer to, and whether those images are available to us for testing.

2. Although the authors presented the results well, I find the inability to integrate all three restorations in one model disappointing. Image datasets might include multiple types of degradations. How is Cellpose3 supposed to be used? Should multiple restorations be used or just one? Although possibly challenging to implement, could there be a preliminary algorithm measuring whether there is and the type of degradation, and would then select the correct restoration? For instance, a CNN classifying images to normal vs. noisy vs. blurred vs. downsampled, and then proceed with the optimal restoration if needed.

Thanks for the feedback. We worked more on this problem and solved it with a modified training protocol. The results are presented in Figure 6, and this new joint model is available in the Cellpose GUI.

Keep in mind that the noise might cover just a small part of the image and not the whole image as in the examples used in this manuscript.

We tested one such scenario and found that the Cellpose3 network can still perform well in such cases. This is presented in new Figure S6.

Minor comments

Title, abstract and introduction

3. The authors have done an excellent job, no critical comments

Results

4. L64-67 “This may be due to sensor noise, which is often described as Gaussian, or shot noise which is well captured by a Poisson distribution.” and L74-77 “Since Poisson noise becomes Gaussian-like in the limit of high simulated brightness, this approach should model both the sensor noise and the shot noise.”

- Please provide evidence of the first sentence and evidence that Poisson noise capture the naturally occurring noise in the samples. Visually, the noise does not reflect the type of noise naturally seen in the images.

This probably depends on what kinds of microscopy methods the reviewer is most familiar with. When the acquisition is photon-limited (such as most fluorescent imaging applications), then the noise is mathematically a Poisson distribution due to the random distribution of photons arriving at the sensor. We have added two citations for this in the new introductory paragraph (Lines 8-9, Meinel et al, 2018 and Taylor and Bowen, 2016). If the reviewer has other types of noise in mind, possibly from other imaging modalities, we would appreciate it if they could describe them, and we could potentially include them in the augmentations.

5. L159-163 “We found that the false positive rates were similar across all denoising approaches and similar to the false positive rate of the original noisy images (Figure S1b-d).

- I agree with the statement when visually examining the plots. Consider providing statistical evidence, for example with p-values of a T-test (or Wilcoxon rank-signed test). Tests are required only for the cellpose3 results vs. the others.

We included this in the caption of the supplementary figures for false positives of cells and nuclei (Figure S2b, Figure S4b). Note that for cells there was no significant difference with the noisy images, and for nuclei the denoising actually reduced the number of false positives.

6. L187-189 “In addition, we trained CARE using pairs of noisy and clean training images from the training data of the specialized dataset”

- Consider adding a reference to the methods, e.g. “See methods” or similar as it is not evident for everyone what CARE is.

Thanks, we added this.

7. L189-190 “Cellpose3 also outperformed this approach.”

- Add mention to the Figure S2b,c.

Thanks, added.

8. L199-201 “We suspect this is because nuclear segmentation relies on simpler object shapes which may be easier to denoise.”

- Do the authors see any possible pitfalls in testing the Cellpose3 in a dataset the Cellpose2 was trained on? Could this also influence that there are less differences between the different losses?

To be clear, the images reserved for testing are always reserved for testing and have not been used for training any of the algorithms (except in the case of Mediar which was trained by a different group using all test images from many datasets).

It is unclear whether *training* Cellpose3 in a dataset where Cellpose2 was trained on would result in problems. The only alternative would be to reserve half the images for training the denoising network with, but that would almost certainly result in suboptimal performance for the segmentation network. We do find that the current approach generalizes well to the test set and to external images: two sets of images from the CARE datasets (Figure 2b,c, 3c,d) and an additional new dataset we collected (Figure 2d,e). It also improves segmentation with Stardist (new Figure S1d). Perhaps this good performance implies that there are no problems with the approach.

9. L223-224 “For deblurring, we trained our networks on images with Gaussian blur of random spatial sizes.”

- Why Gaussian blur as you used Poisson blur before? To my understanding, Gaussian blur does not perfectly reflect naturally occurring blur.

We use a Poisson distribution for adding shot noise independently per pixel, and we use Gaussian kernels for adding spatial blur. These two applications are unrelated to each other, and we motivated above why we used Poisson noise. For spatial blur, the reviewer is correct that Gaussian does not always perfectly reflect natural occurring blur but it should be a good approximation. In some cases the approximation may break, for example for scattering in very deep tissue where adaptive microscopy methods may be required. For such cases we add a suggestion to train new models if the PSF is known, which it often is for these specialized applications using custom microscopes (Lines 269-272).

10. L385 “Each network was trained for 2000 epochs”

- What was the rationale for 2000? Would the results be better by controlling training with checkpoints based on minimizing the diverse loss functions presented in this manuscript?

We had used 500 epochs in the original Cellpose paper, but for these newer models we have a lot more training images so we increased the number of epochs, while keeping the number of images per epoch fixed at 796. We have added this information to the Methods. We have now also included test performance as a function of the number of epochs and the number of training images in the new Figure S1b,c. Our general intuition, confirmed by these analyses, was that the number of iterations has to be “large enough”, and it seems that 2,000 is a value that should almost always be large enough.

11. How well does the restorations work for brightfield images? Could the authors include brightfield (e.g. H&E-stained or giemsa-stained) cell images to demonstrate if the algorithms work with these?

We do have these images in our nuclear dataset and their segmentation is improved by denoising/deblurring/upsampling with Cellpose3. Please see the breakdown by dataset-type in new Figure S8 (MoNuSeg is H&E images).

Discussion

12. In case some of the issues raised above could not be addressed, these should be described in the Discussion part.

Thanks, we think we addressed all the issues with new analyses or discussion items.

Reviewer #3:

Remarks to the Author:

In this manuscript, Carsen et al., proposed an improved strategy for cellular segmentation. In this strategy, the developed method addressed the inherent issues associated with degraded microscopy images, such as noise, blurring, and undersampling. Instead of directly estimating from the noisy images, the authors employed a denoising network to restore the image information, and subsequently utilized a segmentation network to extract cellular information. The approach demonstrated improved performance through a series of simulated experiments and selected real experiments.

However, it is worth noting that the denoising network component of the developed method is derived from existing work, and the segmentation network is based on their previously published work, which suggests a relatively moderate level of novelty in the approach.

The other two reviewers explicitly noted the novelty of Cellpose3. It is possible that the present reviewer missed the complete schema for training Cellpose3: the denoising network is not derived from existing work, but rather it is trained in sequence with the segmentation network to reduce the final segmentation loss. This requires backpropagating gradients through the segmentation network, and then through the denoising network. Thus, the denoising network is specifically trained to produce images that “segment well”. In addition we introduce a perceptual loss, which is novel for biological applications and novel overall for the specific way we use it here. Another novelty is the application of image restoration to a generalist dataset for out-of-the-box generalization to user data, where previously users would have to train their own image restoration models.

Furthermore, the majority of the data utilized in this study involved augmenting existing datasets with noise, blur, and undersampling. Only Fig. 2b and 3c were obtained from actual acquired data.

This was necessary because the only datasets we have found with clean and degraded image pairs (with substantial degradation) were in fact the ones presented in Figures 2b,c and 3c,d. Notice that we did not use this data for training, and we reserved it for testing. The results speak for themselves: even trained “only” on synthetic data, Cellpose3 was able to substantially improve the segmentation quality in these real world datasets. We also acquired a third ground truth dataset in our lab with simultaneous “clean” and “noisy” pairs to further test this and the results hold (Figure 2d,e).

Additionally, the manuscript lacks a comparison with other state-of-art segmentation methods. The author only compared the impact of their own segmentation method with and without denoising, without providing a comprehensive evaluation against other existing approaches.

Since Cellpose3 is primarily an image restoration method, we had compared it with other state-of-the-art image restoration methods like Noise2Void, Noise2Self and CARE. On noisy data, just like Cellpose, other segmentation methods do not work well (new Figure S1a). On Cellpose3-denoised data, other methods can also improve (i.e. StarDist, Figure S1d), but are still outperformed by Cellpose3 overall.

Note that other segmentation methods are in general outperformed by Cellpose even on non-noisy data, and even when trained on the same datasets Cellpose is trained on. For example, in the original Cellpose paper, we found that the Cellpose segmentation network outperforms U-nets and StarDist, and in the Cellpose2 paper, we found that the Cellpose

segmentation network outperforms Mesmer (results also presented in new Figure S9). In this paper, we had included comparisons with a transformer-based version of Cellpose, which did not perform better (Figure 5), and in a separate paper we showed that the winner of the Neurips challenge (Mediar) also does not improve over Cellpose, despite the claims in their paper (Ma et al, Nat Methods 2024).

According to the above consideration, I cannot recommend publication the manuscript.

Besides, there are other minor points to be addressed:

1. Inaccurate label. For example, in Fig. 1c, the authors used 'noise2void' to describe left image. However, the figure caption was described as 'denoised with noise2self (left), and segmented (right)'.

Thanks for catching this, we fixed it.

2. Inconsistent performance. In Fig. 1j, the average precision results showed that there was not a significant difference between using segmentation loss and perceptual + segmentation loss, retrain with noise performed better than using only reconstruction loss. However, in Fig. 3b, the average precision results indicated that the three losses, including reconstruction loss, segmentation loss and perceptual + segmentation loss, showed similar performance. Notably, the reconstruction loss significantly outperformed the retrain with noise approach. This suggests that the performance of using reconstruction loss fluctuates significantly, and there is not a substantial difference between using segmentation loss and perceptual + segmentation loss.

Differences in ordering between the curves in figures 1 and 3 are likely due to differences in the datasets (cells and nuclei, see Lines 223-228). Nuclei are generally easier to segment, and they are easier to denoise as well. As for the benefit of adding a perceptual loss, it is only done to aid in human perceptual recognition, rather than improve the segmentation (Lines 139-151). This may be necessary, for example, when users train new models with the human-in-the-loop approach from Cellpose2, where users have to make their own segmentation decisions. We now added this to the discussion as well (Lines 372-375).

3. In figure 4 (d) on page 5, the average precision curve with perceptual and segmentation loss appears to be the same as the average precision curve with segmentation loss. Why does perceptual loss hardly work for undersampled cellular images? In addition to average precision curves, all experimental results should also be listed in tables.

As mentioned above, the goal of the perceptual loss is to make the images look more interpretable to humans. We added the main results to a table as the reviewer suggested (Supp Table 1).

4. As noted in line 256, individual models are trained for each of the three restoration tasks, and a single model was detrimental to performance. However, real biological images may be simultaneously degraded by noise, blurred, or undersampled.

We have improved our training procedure and can now train a single model that performs well for all tasks (Lines 346-355, Figure 6a-c).

Reviewer #1:

Remarks to the Author:

Thanks to the authors for taking the time to revise the manuscript. I appreciate the addition of the various new validation experiments; the scaling of the training, application to 3D, inhomogeneous blurring, comparison to other segmentation methods with/without cellpose3 denoising, and the revised and improved 'oneclick' denoising model, (Figure 6).

I think the manuscript is greatly improved, addresses the review comments and the method overall now much more convincing and enticing for end-users, particularly with the 'oneclick' restoration model.

I look forward to using Cellpose3 denoising.

Thank you!

Some minor comments of the revised paper:

1) Fig.6b - black line needs labeling in legend + figure for readers. I assume this is the respective base model (cyto2 or nuclei) without denoising.

Thanks for the suggestion, we added labeling in the figure and the legend.

2) Fig. 6d - I found it weird to show visual results for added poisson noise but not for other noise types in the same figure. Is this a typo? as the row label is 'noisy'. If not, it would make sense to also show the other noise types i.e. for deblur and upsampling.

Thanks, we added this.

3) Fig S3C - for completeness and transparency, I recommend including denoiSeg segmentation visualization results so all models plotted in the AP curves is represented.

Thanks, we added this.

4) Fig S9 - the double dashed lines for out-of-distribution is too small in the legend. It looked on first glance identical to the single line. I recommend increasing the size, and including an additional note in the figure legend to clarify, particularly since the 'in-distribution' dataset is different for individual models with the sole exception of cyto3.

Thanks, we fixed this.

Felix Zhou

Reviewer #2:

Remarks to the Author:

The authors addressed all my prior comments and concerns with additional data and analyses.

Thank you!

Reviewer #3:

Remarks to the Author:

Although I am certain that the as-described method will be useful in the cellular segmentation community, I do not think that this work will be as crucial as their two previously published works, either I do not think its importance matches those expected for a journal like Nature Methods.

Some of contributions in the article have already been proposed by existing work, like the combination of restoration and segmentation [1,2], perceptual loss in image restoration [3]. I believe that the authors should and have the ability to pay more attention on the breadth and diversity of training dataset as stated in line 337-339, the core of generalist models.

[1]Paul G, Cardinale J, Sbalzarini I F. Coupling image restoration and segmentation: A generalized linear model/Bregman perspective[J]. International journal of computer vision, 2013, 104: 69-93.

[2]Niu X, Yan B, Tan W, et al. Effective image restoration for semantic segmentation[J]. Neurocomputing, 2020, 374: 100-108.

[3]Zhao H, Gallo O, Frosio I, et al. Loss functions for image restoration with neural networks[J]. IEEE Transactions on computational imaging, 2016, 3(1): 47-57.

**Authors emphasizes that the proposed method is essentially an image restoration approach. However, the comparison with other methods is unfair. For example, the segmentation information used by Cellpose 3 originates from GT images, which definitely including GT content. On the contrary, unsupervised image restoration methods such as Noise2Noise and Noise2self only use the training data itself.

These concerns are quite different from the reviewer's original concerns, which criticized us for not comparing to enough different segmentation methods. To address this, we added more benchmarks to show that other segmentation methods like Mediar and Mesmer do not deal well with degraded images.

However, now they are concerned with our comparisons with other restoration approaches. As described in the title and throughout most of the paper, the goal of Cellpose3 is image restoration specifically for the goal of improved segmentation. This is different from just restoration or just segmentation, and more specific than either, but we think it is a very common use case and worth addressing in itself.

Specifically to their point, it is not true that the other methods "only use the training data itself", because the outputs of all methods including Noise2Noise and Noise2self are segmented with a generalist segmentation algorithm that has been trained on all the GT masks. However, it is true that Cellpose3 can further take advantage of GT masks in ways that other methods cannot. The only exception to this is DenoiSeg, which we had added in the revised manuscript. DenoiSeg however underperforms compared to Cellpose3, and it is based on a very different formulation where segmentation and restoration are merely done with the same neural network, rather than restoration being trained specifically for the goal of segmentation. This crucial distinction is what makes our method novel, as appreciated by the other reviewers. The references 1&2 that the reviewer offers are essentially DenoiSeg but for general computer vision problems: the denoising and segmentation are merely formulated under the same joint optimization criterion, rather than denoising being optimized as a step towards improving segmentation. As we can see from the results, this distinction really does matter.

It has been the goal of all Cellpose papers to do new kinds of tasks that hadn't been done before. As such, it is usually hard to find approaches that solve the exact same problem using

the same exact training data. In this case and with the help of the reviewers, we think we've done the best we could identifying DenoiSeg, Noise2Noise, Noise2Self and CARE as alternative tools people might use to solve these kinds of problems.

In practical scenarios, microscopic images often require manual screening or localization of degraded areas, where many degradations are scattered and localized. As a "one click" model, the authors did not present the results of a real and complete image, nor did they explain the impact of clear images passing through the network.

We addressed this concern with the partial blurring experiments (Figure S6), which we performed in the first round of revisions.